# DIMENSION DOMAIN CO-DECOMPOSITION: SOLVING PDES WITH INTERPRETABILITY

## ABSTRACT

Physics-informed neural networks (PINNs) have shown promise for solving partial differential equations (PDEs), but they face significant challenges in high-dimensional settings and when modeling solutions with sharp features. Existing approaches also lack interpretable per-dimension representations and depend on manually defined domain partitions. To address these challenges, we propose a unified Dimension Domain Co-Decomposition (3D) framework that integrates dimension decomposition with a Mixture-of-Experts (MoE) based domain decomposition. Our approach achieves three key innovations. First, we introduce an interpretable dimension decomposition strategy that decouples individual coordinate inputs within each expert using a single shared MLP with indexed inputs, significantly reducing the model size. Second, we propose a novel metric, Variable Interpretability ($VI$), that quantifies the alignment between the learned latent representations of each input dimension and their corresponding exact solution components. Third, we present an MoE-driven domain decomposition architecture that automatically partitions the solution space without requiring predefined regions or interface conditions. Extensive experiments demonstrate that our approach improves both computational efficiency and solution accuracy across a range of high-dimensional PDE benchmarks, with interpretable and scalable performance.

## 1 INTRODUCTION

Partial differential equations (PDEs) provide the mathematical foundation for describing a wide range of physical and engineering phenomena, including fluid dynamics Anderson, 1995, wave propagation (Strauss, 2007), and quantum mechanics (Griffiths & Schroeter, 2018). Classical numerical solvers such as the finite element method (FEM) (Zienkiewicz et al., 2005; Babuška, 1971), the finite difference method (FDM) (LeVeque, 2007; Lax & Richtmyer, 1956), and the spectral method (SM) (Trefethen, 2000; Boyd, 2001) have long been the standard tools for approximating PDE solutions. FEM is flexible for handling irregular domains, FDM is simple and efficient on structured grids, while SM achieves spectral (fast) convergence but is restricted to periodic boundary conditions. Despite their success, both methods suffer from rapidly increasing computational cost when dealing with high-dimensional problems, complex nonlinearities, or solutions with sharp local features, which often makes them impractical for large-scale applications.

In recent years, neural networks have emerged as promising alternatives for PDE solving, either by directly approximating solutions from data or by embedding the governing equations into the training objective through physics-informed neural networks (PINNs) (Raissi et al., 2019). PINNs offers clear advantages in high-dimensional settings where traditional numerical solvers become infeasible. Building on this flexibility, two major lines of decomposition-based methods have been explored to further enhance scalability and adaptivity. Dimension decomposition improves scalability by factorizing solutions along coordinates (Cho et al., 2023; Liu et al., 2024). This strategy simplifies optimization in high-dimensional settings and further mitigates the curse of dimensionality. However, existing approaches lack interpretability measurement. In contrast, domain decomposition (Jagtap et al., 2020a; Shukla et al., 2021; Hu et al., 2023) focuses on local adaptivity by dividing the computational domain into smaller subdomains, with each subdomain handled by a specialized model. This enables better approximation of both smooth and discontinuous solutions. Nevertheless, such methods typically rely on manually pre-defining the subdomains. When

the subdomains overlap, one must introduce extra loss terms to ensure the predictions agree in the overlapping regions; when the subdomains are non-overlapping, additional conditions are required to enforce continuity across the shared boundaries. These constraints make the training procedure more complicated and problem-dependent.

To overcome these limitations, we propose Dimension Domain Co-Decomposition (3D), a unified framework that combines both decomposition strategies in a scalable, interpretable, and fully automatic manner. At the dimension level, each variable is modeled separately, which improves scalability in high dimensions. In practice, these dimension components are processed through a shared MLP, ensuring parameter efficiency across coordinates. At the domain level, 3D employs Mixture of Experts (MoE) (Jacobs et al., 1991). It contains multiple experts, and a router assigning soft weights to combine their outputs. This mechanism encourages each expert to specialize in certain subregions, so that domain decomposition emerges adaptively during training. As a result, 3D can effectively capture solutions with sharp local features without requiring pre-defined regions or explicit interface conditions. An illustration of 3D with two experts on input $[t, x]$ is shown in Figure 1. In addition, we propose Variable Interpretability ($VI$), a quantitative metric that matches predicted per-dimension components to ground-truth factors. $VI$ takes values in $[0, 1]$, with 1 indicating perfect alignment across variables.

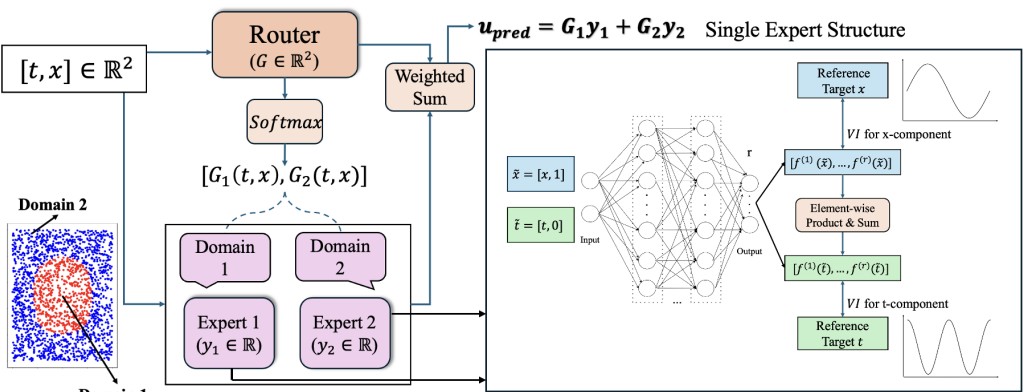

Figure 1: **Structure of 3D (Dimension Domain Co-Decomposition) with two experts.** *Left (Mixture-of-Experts).* The router takes the spatiotemporal input $[t, x]$ and produces two gating weights after a softmax. These weights induce an automatic partition of the domain (illustrated by the red/blue subdomains). The model's prediction is obtained as a weighted sum of the two expert outputs $y_1, y_2$. *Right (Expert structure).* Each expert takes the same input $[t, x]$ and feeds two indexed streams (one for $x$ and one for $t$) into a shared MLP, which produces $r$ latent components for each. The $x$-$t$ component pairs are combined by element-wise multiplication, and then summed over the $r$ pairs to yield the expert's output. Reference Target $x$ and Reference Target $t$ is used to compute $VI$ by comparing them with the learned $x$ and $t$ components. Together, the router and experts realize domain decomposition and dimension decommposition within each subdomain.

We summarize our contribution as follows:

- We propose Dimension Domain Co-Decomposition (3D), a unified framework integrating dimension decomposition with adaptive domain decomposition for solving high-dimensional PDEs.

- Within 3D, we design a lightweight shared-MLP architecture that processes dimension-index pairs, enabling reduced model size while capturing coordinate-wise features.

- We introduce Variable Interpretability ($VI$), a novel, quantitative, scale-invariant metric to evaluate dimension-wise interpretability. It evaluates the alignment between the learned latent representation of each input dimension and the ground-truth components, thereby serving as a direct measure of interpretability.

- We employ MoE to induce an adaptive and automatic domain decomposition capturing sharp features without requiring predefined subdomains or explicit interface conditions.

## 2 RELATED WORK

### 2.1 DIMENSION DECOMPOSITION AND INTERPRETABILITY

High-dimensional PDEs pose significant challenges for neural network-based solvers. Building on the PINNs framework, several recent works (Cho et al., 2023; Liu et al., 2024; Vemuri et al., 2024; Liu et al., 2022) introduce dimension-wise decomposition strategies to mitigate the curse of dimensionality. Most of these approaches rely on classical tensor decomposition techniques (Tucker, 1966; Carroll & Chang, 1970), which improve efficiency by reducing the representation complexity, but still assign a separate neural network to each dimension, leading to suboptimal efficiency. At the same time, these methods offer little interpretability measurements of the learned components. In parallel, the interpretable machine learning community has developed models such as GAMs, NAMs, and self-explaining networks (Hastie & Tibshirani, 1990; Wood, 2017; Agarwal et al., 2021; Alvarez-Melis & Jaakkola, 2018; Lou et al., 2013), which represent the target function as a sum of univariate functions, each depending on a single variable. These models offer intuitive per-variable explanations, but their additive structure struggles to capture higher-order interactions, which are often intrinsic to PDE solutions. Beyond additive models, sparse regression–based methods such as SINDy and its variants (Brunton et al., 2016; Kaiser et al., 2018) provide another line of interpretability by discovering governing equations from data. Unlike variable-wise interpretability, these methods explain the underlying physical laws by identifying symbolic equations, rather than uncovering the structures of PDE solutions themselves. To fill in these gaps, We propose a shared-MLP dimension decomposition that removes redundant per-dimension networks for greater efficiency, and introduce Variable Interpretability ($VI$), a metric quantifying the alignment between learned components and ground-truth factors.

### 2.2 DOMAIN DECOMPOSITION OF PINNS

Domain decomposition has been widely adopted to improve PINNs for solving complex PDEs. The XPINNs framework (Jagtap et al., 2020c) pioneered this idea by partitioning the computational domain into multiple subdomains and training a separate PINN in each region; to ensure consistency, XPINNs enforces continuity of the solution across subdomain interfaces through additional interface losses. Subsequent works have refined this approach: Shukla et al. (Shukla et al., 2021) introduced parallel implementations combining cPINNs (Jagtap et al., 2020b) and XPINNs, exploiting overlapping Schwarz-type decompositions to better handle multi-scale problems. Hu et al. (2023) proposed APINNs, which use soft gating mechanisms to allow more flexible domain decomposition. Dolean et al. (Dolean et al., 2024) developed multilevel decomposition architectures to improve accuracy for large or highly heterogeneous domains. More recently, the approach named BPINN (Vicens Figueres et al., 2025) integrates Bayesian PINNs with domain decomposition, computing local uncertainties concurrently and enforcing interface flux continuity among subdomains. There are also specialized applications, such as domain decomposition PINNs for incompressible Navier–Stokes equations (Gu et al., 2024). Despite these advances, a common limitation is that all existing approaches require predefined partitions of the computational domain. Moreover, additional conditions must be imposed at the subdomain interfaces to guarantee continuity of the solution. These requirements restrict adaptivity and limit the flexibility of domain decomposition when applied to PDEs with unknown or heterogeneous solution structures. In contrast, our framework enables automatic and adaptive domain decomposition during training.

## 3 DIMENSION DOMAIN CO-DECOMPOSITION

Existing PINNs-based methods for high-dimensional PDEs suffer from three obstacles: (i) high computational cost due to the need for dense collocation sampling; (ii) a lack of principled interpretability metric for dimension-wise factorizations, where scaling, permutation, and cross-dimension mixing obscure whether learned components reflect the underlying physics; and (iii) brittle domain decomposition that depends on predefined subdomains and delicately tuned interface penalties, making performance sensitive to the chosen partition and enforcement strength. To address these issues, we adopt a dimension decomposition that reduces computation-graph complexity by combining them in a low-rank manner; we introduce Variable Interpretability ($VI$) to quantify alignment between learned per-dimension components and reference factors; and we develop MoE-driven domain decomposition that maps the input coordinates to soft expert assignments, avoiding

manual region design and explicit interface enforcement. In combination, the dimension decomposition lowers training cost, $VI$ provides quantitative interpretability, and the MoE router delivers robust, automatic domain partitioning. Given input $\mathbf{x} = [x_1, x_2, \cdots, x_d]$, the predicted solution $\hat{u}$ takes the form:

$$\hat{u}(x_1, x_2, \cdots, x_d) = \sum_{i=1}^{K} G^{(i)}(\mathbf{x}) E_i(\mathbf{x}), \ E_i(\mathbf{x}) = E_i(f_1(x_1), f_2(x_2), \cdots, f_d(x_d)) \qquad (1)$$

where $f_j$ for $j = 1, 2, \cdots, d$ stands for the Multilayer Perceptron (MLP) processing each dimension component. $E_i$ for $i = 1, 2, \cdots, K$ represents expert while $G(\mathbf{x}) \in \mathbb{R}^K$ is a router assigning weights for experts.

In section 3.1, we present the structure of a single expert and explain its role in achieving dimension decomposition. Section 3.2 introduces the $VI$ for assessing dimension interpretability. Section 3.3 describes the overall MoE-driven domain decomposition framework.

### 3.1 DIMENSION DECOMPOSITION IN 3D FRAMEWORK

Conventional methods mix all dimensions in a single network. For high-dimension problems, large number of data complicates the computation graph, making both forward and, more severely, backward propagation expensive. We adopt dimension decomposition in single expert to decouple coordinates and simplify both forward propagation and derivative computation. Our domain decomposition design is similar in form to the Canonical Polyadic Decomposition (CP-decomposition) (Carroll & Chang, 1970; Harshman, 1970). Conventionally, for $d$-dimensional input, the output can be written as follows:

$$\hat{u}(x_1, ...x_d) = \sum_{i=1}^{r} f_1^{(i)}(x_1) f_2^{(i)}(x_2) \cdots f_d^{(i)}(x_d) \qquad (2)$$

where $\hat{u} : \mathbb{R}^d \to \mathbb{R}$ is the predicted solution, $x_j \in \mathbb{R}$ is a coordinate of $j$-th component including temporal coordinates if exist. $f_j(x_j) : \mathbb{R} \to \mathbb{R}^r$ represents independent MLP processing $x_j$. $r$ is comparable to the rank in CP-decomposition. In our settings, $r$ impacts more on Variable Interpretability ($VI$) than accuracy. Modest $r$ are sufficient-typically $r \in \{4, \cdots, 16\}$ achieving good interpretability while maintaining satisfactory accuracy, see section 4.

However, independent per-axis processing introduces a large number of parameters. We address this issue by using a single shared MLP to model all dimension components within each expert. Specifically, each component is represented by a two-dimensional input vector consisting of the coordinate value and its index. For the $j$-th dimension component, the corresponding output is given by $f(x_j, , j - 1)$. For example, for 3d PDE problem, outputs of dimension components are $f(x_1, 0), f(x_2, 1), f(x_3, 2)$. We treat temporal coordinate $t$ as part of the physical vector coordinates. Therefore, equation 2 can be rewritten into:

$$\hat{u}(x_1, ...x_d) = \sum_{i=1}^{r} f^{(i)}(x_1, 0) f^{(i)}(x_2, 1) \cdots f^{(i)}(x_d, d - 1) \qquad (3)$$

Our framework bases on PINNs. Therefore, the loss function can be written as follows:

$$Loss = w_{pde} Loss_{pde} + w_{ic} Loss_{ic} + w_{bc} Loss_{bc} \qquad (4)$$

where $Loss_{pde}$ is the PDE residual loss, which penalizes the discrepancy between the neural network prediction substituted into the PDE and the equation's right-hand side at sampled collocation points. $Loss_{ic}$ and $Loss_{bc}$ represent initial-condition loss (for time dependent problems) and boundary-condition loss, respectively. More information is included in Appendix B.

The proposed architecture is related to Separable Physics-Informed Neural Networks (SPINNs) (Cho et al., 2023), but it differs in several key aspects that bring advantages: First, we use single MLP processing each dimension component with an additional index as input, saving the memory when handling high-dimensional problems, see section 4.2 for more information. Second, our framework naturally integrates with a MoE structure. While SPINNs rely on forward-mode automatic differentiation (AD), this is not directly compatible with MoE because the router breaks the

separable structure. Instead, we adopt reverse-mode AD which allows the decomposition to remain effective while benefiting from adaptive domain specialization. Lastly, the dimension decomposition design enables us to bypass meshgrid collocation points. Instead of constructing a full grid, we independently sample training points for each dimension component and then combine them, which drastically reduces the number of collocation points required and improves training efficiency.

## 3.2 VARIABLE INTERPRETABILITY (VI)

Previous dimension decomposition techniques lack quantitative interpretability for dimension component. To address this gap, we propose a new metric that evaluates each dimension component by comparing it against the reference target (either analytical or high-accuracy numerical). Concretely, for $j$-th dimension component, we obtain $f(x_j, j-1) \in \mathbb{R}^r$ from dimension decomposition. Evaluating this function on $n_j$ sampled points produces $n_j$ row vectors in $\mathbb{R}^r$, which we stack to form a matrix $F_j \in \mathbb{M}^{n_j \times r}$. In parallel, we construct the ground-truth matrix $G_j \in \mathbb{M}^{n_j \times s}$ by evaluating the exact $j$-th factor at the same points. For example, in the 5D Poisson equation with solution $u(x) = \prod_{j=1}^{5} \sin(\pi x_j)$, the predicted $x_j$-component is represented as $F_j \in \mathbb{M}^{n_j \times r}$, while the ground-truth factor is $g_j(x) = \sin(\pi x_j)$, evaluated on $n_j$ points to form $G_j \in \mathbb{M}^{n_j \times 1}$. For simplicity, we use $F$ and $G$ in the remainder of this section.

Before computing the metric, both $F$ and $G$ are normalized. Take $F$ as an example:

$$\tilde{F}_{ik} = \frac{F_{ik} - \mu_k}{\max(\sqrt{\sum_{q=1}^{n}(F_{qk} - \mu_k)^2}, \epsilon)}, i = 1, ..., n, k = 1, ..., r \tag{5}$$

where $\mu_k = \frac{1}{n}\sum_{q=1}^{n} F_{qk}$, $\epsilon = 10^{-12}$ to avoid denominator is 0.

Then we apply the QR decomposition to $\tilde{F}$ and $\tilde{G}$ to obtain the reduced orthonormal bases $Q_{\tilde{F}}$ and $Q_{\tilde{G}}$. We then compute the singular values $\{\sigma_i\}_{i=1}^{m}$ of $Q_{\tilde{F}}^{\top} Q_{\tilde{G}}$, where $m = \min(\text{rank}(Q_{\tilde{F}}), \text{rank}(Q_{\tilde{G}}))$. The $VI$ of the $j$-th component is defined as:

$$VI_j = \frac{1}{m}\sum_{i=1}^{m} \sigma_i^2. \tag{6}$$

$VI_j$ takes values in the range $[0, 1]$ with values closer to 1 indicating a better fit to the exact terms. For each problem, we then take the mean of $VI_j$ across $j$ to get final $VI$.

Notably, this metric evaluates all-rank representation features as a whole, testing how well the subspace spanned by the exact basis $Q_G$ is aligned with (and contained in) the subspace spanned by the predicted basis $Q_F$. In practice, the exact matrix $G$ often has shape $(n, s)$ with $s \leq r$, where $r$ is the decomposition rank. Thus, the number of exact basis vectors can be smaller than the number of predicted ones. Only when $s = r$, $VI = 1$ means the predicted subspace and the exact subspace are identical. For example, $G_x \in \mathbb{M}^{n \times 1}$ in 5d Poisson equation while we use $r > 1$ in section 4.2. In this way, $VI = 1$ means that the exact one-dimensional subspace is fully contained in the predicted subspace. In short, when $s < r$, $VI$ measures whether the predicted subspace totally covers the exact subspace instead of testing if two subspaces are identical. This case is particularly relevant in practice, since $r$ can be chosen arbitrarily large while the number of exact basis vectors $s$ is often much smaller.

## 3.3 MOE-DRIVEN DOMAIN DECOMPOSITION

Partitioning the solution domain into subdomains enables local specialization of the underlying physics, improving accuracy and stability. Previous domain decomposition methods require manually pre-defined regions and interface conditions. To achieve automatic and adaptive domain decomposition, we adopt a Dense MoE model (Jacobs et al., 1991). Compared with Sparse MoE (Shazeer et al., 2017), dense MoE avoids expert collapse and provides more stable training. This is important in problems with shocks where top-$k$ gating may cause instability near shocks. Router is a MLP $G: \mathbb{R}^d \to \mathbb{R}^K$ taking only $\mathbf{x} \in \mathbb{R}^d$ (including temporal and spatial coordinates) as input. It produces logits which are then converted into mixture weights via a softmax. The weight assignment serves as a soft partition indicator – large weight marks the region where expert is responsible for. Each

expert $E_i$ for $i = 1, 2, \cdots, K$ specializes in local regions. It remains smooth within its responsible region while differing from other experts to cover complementary behaviors. Together, they provide global approximation by $\sum_{i=1}^{K} G(\mathbf{x}) E_i(\mathbf{x})$.

Since the predicted solution is the weighted sum of experts' outputs, the overall loss function follows equation 4, except for the computation of $\hat{u}$. All experts share same architectures and inputs with separate parameters. End-to-end training is performed. Both the router and experts are updated via gradient descent optimization. Our experiment results demonstrate that increasing the number of experts $K$ initially leads to significant error reduction and reflects finer domain decomposition. However, beyond a certain number $K_{optimal}$, additional experts yield similar errors and no more new information about domain decomposition. In practice, we select $K_{optimal}$ as best number of experts.

## 4 EXPERIMENTS

### 4.1 EXPERIMENT SETUP

We evaluate our framework on two settings: (i) Dimension decomposition (mainly Poisson and Wave equations), and (ii) MoE-driven domain decomposition in which each expert uses the same dimension decomposition architecture (Viscous Burgers and Linear Transport equations). In our experiments, training is first performed with the Adam optimizer for fast convergence and followed by LBFGS for refinement. A cosine-annealing scheduler is applied to adjust the learning rate. Training performances are measured by the relative $\ell_2$ error. All experiments are trained on a single NVIDIA RTX 5090 GPU.

**Dimension Decomposition and Interpretability.**  Our framework is built on a unified expert design, where each expert employs a shared MLP for dimension decomposition. These experts are either combined under a MoE structure (Viscous Burgers and Linear Transport) or used as a single module (Poisson and Wave equations). To evaluate the scalability and efficiency of this shared MLP design, we first conduct a parameter count comparison across all four PDE benchmarks. In the subsequent studies, we focus on Poisson and Wave equations with a single expert module to highlight the effect of dimension decomposition and quantify interpretability using the proposed $VI$. The shared MLP within each expert module consists of two hidden layers of width 64 with Tanh activation by default.

**Domain Decomposition.**  Viscous Burgers equation and Linear Transport equation (Appendix A) employ MoE-driven domain decomposition while keeping dimension decomposition inside each expert, showing not only domain decomposition but also comprehensive test of 3D framework. Dense MoE with multiple experts are applied. The router was set to be a 5-layer MLP with width 64 per layer and Tanh activation.

### 4.2 DIMENSION DECOMPOSITION AND INTERPRETABILITY

**Benefit of Shared MLP.**  We first demonstrate the benefit of the shared MLP inside each expert module. Table 1 compares the number of trainable parameters across different PDE problems. We fix $r = 16$ for this parameter test. For Poisson, Wave equations, we adopt a single expert. For the Linear Transport and Viscous Burgers equation, we use 3 experts and 2 experts, respectively. Across all settings, the shared MLP design significantly reduces the number of trainable parameters compared with independent MLPs design. The advantage enlarges as the input dimension grows, highlighting the scalability of our approach. In the context of a single expert module, the parameter count of a shared MLP is independent of the input dimension, whereas it grows with the dimension for independent MLPs. When extended to a MoE framework, the shared MLP architecture reduces the overall number of parameters by sharing them across the experts.

Shared MLP architecture also leads to reduced memory. Generally, the shared design requires on average 77.8% of the memory compared to independent MLPs. The efficiency gain scales with dimensionality: in the 5d Poisson problem, the shared MLP reduces memory consumption to 50.0%, and in the 10d Poisson problem, the shared design achieves an even greater reduction, using only 30.4% of the memory.

Table 1: Comparison of number of trainable parameters between shared MLP and independent MLPs design.

| Type | 5d Poisson | 10d Poisson | 1d Wave | 2d Wave | Burgers | Transport |
|------|------------|-------------|---------|---------|---------|-----------|
| Shared MLP | 5392 | 5392 | 5392 | 5392 | 23586 | 29043 |
| Independent MLPs | 26640 | 53280 | 10656 | 15984 | 34114 | 44835 |

The training performance of the shared MLP is comparable to that of independent MLPs with far fewer parameters, and both clearly outperform vanilla PINNs (Figure 2). For vanilla PINNs, we adopt a 10-layer MLP with width 64 and Tanh activation. We report results on the 5d Poisson equation. Since training stops once the convergence condition is met, the total number of training steps varies across models. For comparison, we truncate at the smallest step count, 11,400, which corresponds to the termination of both the shared and independent MLPs. By contrast, vanilla PINNs stop at 23,400 steps. At termination, the shared MLP, independent MLPs, and vanilla PINNs achieve $\ell_2$ errors of $1.8430 \times 10^{-4}$, $3.2620 \times 10^{-4}$, and $7.5451 \times 10^{-3}$, respectively.

Furthermore, we evaluated the training performance on the 10d Poisson problem. For fairness, the baseline PINNs uses a single MLP with four hidden layers and width 64, identical to the shared MLP configuration. With a comparable number of parameters (5392 for the shared MLP versus 4929 for the baseline PINNs), the shared MLP with rank $r = 16$ achieves a relative $\ell_2$ error of $1.25 \times 10^{-3}$ after only 11,500 epochs. In contrast, the standard PINN requires 31,500 epochs yet converges to a much worse error of $1.29 \times 10^{-1}$. Although the shared-MLP requires a bit higher per-epoch cost, resulting in a total training time of 1579 s versus 1184 s, the substantial gain in accuracy outweighs this moderate runtime trade-off. These results indicate that the shared-MLP provides a far more expressive representation for high-dimensional Poisson problems, even under comparable model capacity.

Moreover, the separable parameterization supports dimension expansion: a model trained in a lower-dimensional setting can be directly fine-tuned to higher-dimensional problems, whereas standard MLP-based PINNs cannot be reused due to mismatched input dimensionality. We fine-tuned a 5D model on the 8D Poisson problem, accelerating convergence and achieving better accuracy. Complete fine-tuning details and results are provided in the **Appendix C**.

**Interpretability.** We train the Poisson equation with 8192 collocation points and 2048 boundary points using an expert module. The values in Table 2 represent the mean $VI$ averaged over five independent random seeds in all the dimensions. From the analytical solution $u = \prod_{i=1}^{5} \sin(\pi x_i)$ (Appendix A), one might expect that $r = 1$ would suffice for interpretability compared to equation 3. However, our experiments demonstrate that $r = 1$ is insufficient. By increasing $r$ to 4, we obtain $VI \approx 1$, as reported in Table 2. For higher dimensions, we further test the 10d Poisson problem. Even in this case, full interpretability ($VI = 1$) is achieved with $r = 5$, and the model also attains a satisfactory accuracy with $\ell_2$ error $0.0025 \pm 0.0028$. These results confirm that a small value of $r$ ensures good interpretability while maintaining strong learning performance. For completeness, we further evaluate our framework on a 2d Poisson equation defined on an L-shaped

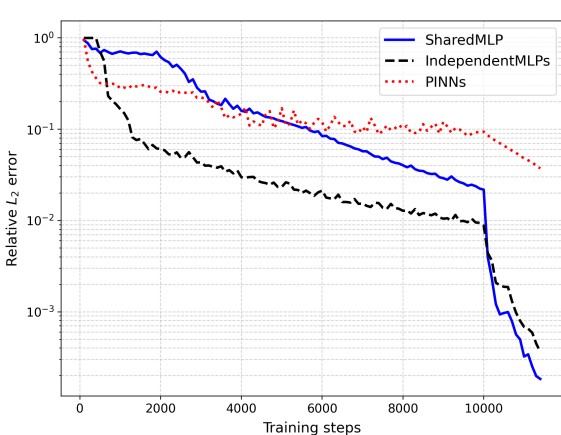

Figure 2: **Comparison of relative $\ell_2$ error (log scale) for 5d Poisson between shared MLP, independent MLPs and vanilla PINNs.** Training is displayed up to 11,400 steps, where both the shared MLP and independent MLPs converge. The final $\ell_2$ errors are $1.8430 \times 10^{-4}$ (shared MLP), $3.2620 \times 10^{-4}$ (independent MLPs), and $7.5451 \times 10^{-3}$ (vanilla PINNs).

domain, demonstrating that the method extends naturally to irregular geometries. The full setup and results are provided in Appendix C.

Table 2: Mean $VI$ over dimension components for different values of $r$ across the PDE examples. All values are averaged over five independent random seeds and are reported in percentage form (i.e., multiplied by 100).

| PDE examples | r=1 | r=2 | r=3 | r=4 | r=5 |
|---|---|---|---|---|---|
| 5d Poisson | $4.11 \pm 0.00$ | $91.21 \pm 12.66$ | $99.72 \pm 0.14$ | $99.99 \pm 0.01$ | $100.00 \pm 0.00$ |
| 10d Poisson | $4.82 \pm 1.10$ | $87.48 \pm 7.49$ | $99.46 \pm 0.06$ | $99.99 \pm 0.01$ | $100.00 \pm 0.00$ |
| 1d Wave $c = 2$ | $100.00 \pm 0.00$ | $100.00 \pm 0.00$ | $100.00 \pm 0.000$ | $100.00 \pm 0.00$ | $100.00 \pm 0.00$ |
| 1d Wave $c = 5$ | $49.26 \pm 1.04$ | $83.09 \pm 2.82$ | $90.65 \pm 6.78$ | $90.72 \pm 6.64$ | $99.40 \pm 0.10$ |
| 1d Wave $c = 10$ | $41.71 \pm 8.02$ | $54.71 \pm 10.56$ | $58.39 \pm 3.58$ | $59.23 \pm 2.83$ | $84.59 \pm 3.42$ |
| 2d Wave $c = 2$ | $67.56 \pm 1.34$ | $94.53 \pm 3.10$ | $99.74 \pm 0.19$ | $99.97 \pm 0.02$ | $100.00 \pm 0.00$ |

For the Wave equation, we use 8192 collocation points, 1024 initial points, and 1024 boundary points, again with a single expert module. In this setting, $r = 1$ is sufficient to achieve full interpretability with $VI = 1$, consistent with the analytical solution $u(t, x) = \sin(\pi x) \cos(c\pi t)$ (Appendix A). We first examine the 1d case with $c = 2$. Figure 3 compares the predicted and exact components $f_t(t) = \cos(c\pi t)$ and $f_x(x) = \sin(\pi x)$ at training steps 1000, 2000, 3000, and 4000. As suggested by the analytical solution, the $t$-component has a higher frequency than the $x$-component. Accordingly, the model learns $f_x(x)$ within the first 1000 steps but requires up to 4000 steps to accurately capture $f_t(t)$. This behavior is fully consistent with a well-known limitation of PINNs: higher-frequency structures are intrinsically harder for PINNs to learn, often requiring more optimization steps and finer resolution. We then test cases with $c = 5$ and $c = 10$, where the solution includes higher-frequency terms $\cos(c\pi t)$. In these settings, $r = 1$ is no longer sufficient for full interpretability. Nevertheless, $VI$ improves as $r$ increases, reaching $VI \approx 1$ for $c = 5$. Finally, we consider the 2d Wave equation with analytical solution $u(t, x_1, x_2) = \sin(\pi x_1) \sin(\pi x_2) \cos(\sqrt{2}c\pi t)$ and $c = 2.0$. The additional spatial dimension increases the learning difficulty, as reflected in Table 2. Still, the model achieves $VI = 1$ at $r = 5$, underscoring that small values of $r$ suffice to ensure strong interpretability.

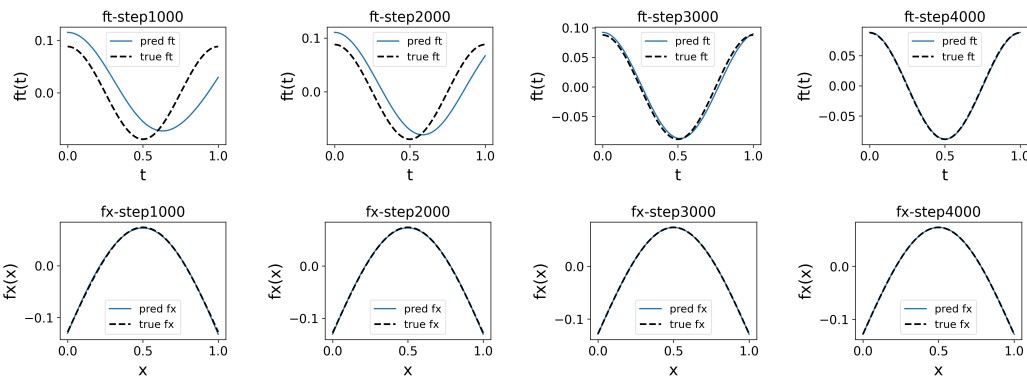

Figure 3: **Components' interpretability of 1d Wave equation when $c = 2$.** The first row represents comparison of $t$-component while the second row represents comparison of $x$-component. Here "pred $f_x$" and "pred $f_t$" in the figure refers to the shared MLP processing $x$ and $t$ respectively. The black dotted line stands for true value from analytical solution and the blue solid line stands for predicted value. From left to right, the columns represent the 1000th, 2000th, 3000th, and 4000th training steps, respectively.

### 4.3 DOMAIN DECOMPOSITION

Multiple experts and a router are employed for automatic domain decomposition for Viscous Burgers. Within each expert, dimension decomposition is applied. The shared MLP consists of two hidden layers of width 32 with $r = 16$. The training data consists of 10,000 randomly sampled collocation points, 256 initial points, and 200 boundary points. For testing, we adopt high-accuracy dataset generated in MATLAB, as in PINNs (Raissi et al., 2019).

For the tested viscosity $\nu = \frac{0.01}{\pi}$ (see Appendix A), the shock at $x = 0$ represents the solution's main discontinuity. It is, therefore, the natural choice for the splitting boundary in domain decom-

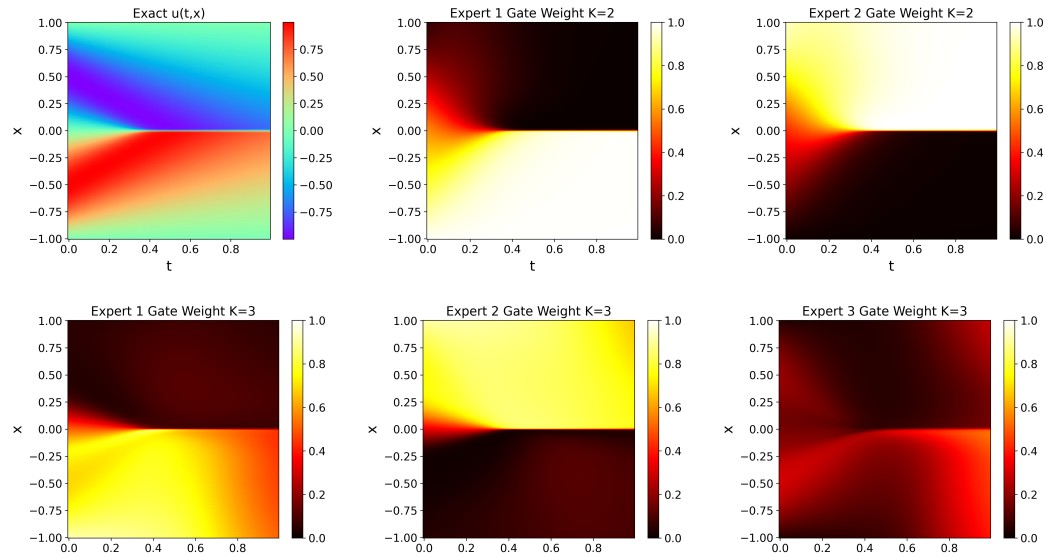

Figure 4: **Ground truth and domain decomposition results of Viscous Burger for $K = 2$ and $K = 3$.** The left panel in the first row shows the ground truth solution. The remaining two panels in the first row display the decomposition results with $K = 2$, indicating obvious boundary of $x = 0$. Three figures in the second row correspond to $K = 3$, which shows little new decomposition information. For $K = 1, 2, 3$, $\ell_2$ error achieves $0.2108 \pm 0.1252, 0.0011 \pm 0.0005, 0.0008 \pm 0.0004$, showing effectiveness of MoE structure.

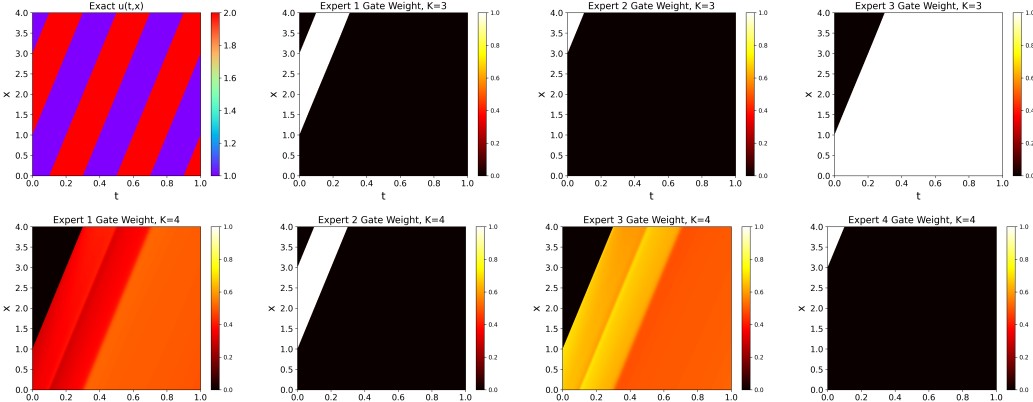

Figure 5: **Results of domain decomposition of Linear Transport for $K = 3, 4$.** The left panel in the first row shows the ground truth solution. The remaining three panels illustrate weight assignments of experts when $K = 3$, demonstrating clear cut-off lines same as the ground truth. Four panels in the second row are domain decomposition results of $K = 4$, displaying more detailed partition.

position (see Figure 4). Although the number of experts affects the precise partitioning, this critical shock location can be consistently identified. We visualize the router assignments for each expert. Figure 4 shows the domain decomposition results for Viscous Burgers with different numbers of experts $K$. It is evident that for $K = 2$, the model achieves domain decomposition, mainly separated by shock at $x = 0$. Increasing the number of experts to $K = 3$ does not introduce new meaningful subdomains since the additional expert tends to receive small weights. For $K = 1, 2, 3$, $\ell_2$ error achieves $0.2108 \pm 0.1252, 0.0011 \pm 0.0005, 0.0008 \pm 0.0004$, showing effectiveness of MoE structure. Appendix C provides additional visualizations for $K = 4$ and $K = 5$ and an ablation analysis of how $r$ affects the error.

We further evaluate the 1d Linear Transport equation. In the main paper, we present results for the case with clearly separable regions (Appendix A), while the case with smooth transitions is deferred

to Appendix C. For both settings, we use 8192 collocation points along with 1024 initial points and 1024 boundary points.

For MoE-driven domain decomposition, we find that using three experts ($K = 3$) yields a reasonable decomposition. In this case, we set $r = 4$. The learned partition successfully captures the diagonal stripe structures observed in the ground truth (Figure 5), with the predicted stripe locations closely matching those of the exact solution. Results with four experts ($K = 4$) are shown in the second row of Figure 5, where we use $r = 8$. The additional expert produces a more detailed partition. Further results for other values of $K$ are provided in Appendix C, demonstrating that too few experts lead to unclear decompositions, while larger $K$ do not yield additional structural information.

**Consistency.** To evaluate the consistency of the learned domain decompositions, we repeat the Viscous Burgers and Linear Transport problems under five different random seeds that control the random initialization of all network parameters. We fix collocation points across runs. Across different seeds, the MoE-based domain decompositions recover the prominent structures: in Viscous Burgers, the shock location at $x = 0$ is consistently distinguished, while in the Linear Transport problem, different experts align with the diagonal stripe patterns. Representative visualizations under different seeds are provided in Appendix C. This shows that the domain decomposition is driven by intrinsic geometric features of the PDE solutions.

**Robustness.** We further test robustness by adding relative Gaussian noise up to $5\%$ to the initial and boundary conditions. The resulting MoE-based domain decompositions remain stable. Visualizations comparing the noise-free and noisy settings are provided in Appendix C.

## 5 CONCLUSION

In this paper, we propose Dimension Domain Co-Decomposition (3D), a PINNs-based framework that unifies dimension decomposition and MoE-driven domain decomposition. Within each expert, a shared MLP processes coordinate–index pairs to produce dimension-wise functions. To quantify the alignment between predicted dimension component and ground truth component, we introduce Variable Interpretability ($VI$). At the MoE level, the router adaptively partitions the domain so that experts specialize in local regions without requiring predefined subdomains or explicit interface conditions. Through experiments on PDE benchmarks, we show that 3D not only achieves good accuracy but also produces interpretable decompositions across dimensions according to $VI$. Nevertheless, our study has limitations. $VI$ relies on reference solutions that are dimension-separable. For non-separable solutions, we must construct separable approximations, for example using truncated Fourier series, and compare the predicted components against these numerical factors. Future work should explore more general interpretability metrics that extend beyond separable settings.

## REPRODUCIBILITY STATEMENT

We are committed to ensuring the reproducibility of our results. All code for our framework (training, evaluation, and visualization) is attached as supplementary material. For clarity, each PDE example (Poisson, Wave, Burgers, Transport) is implemented in a separate code file named after the corresponding PDE problem, making it straightforward to reproduce individual experiments. PDE datasets are generated from analytic or high-accurate numerical solutions as described in Section 4.2. We provide all hyperparameter settings in Appendix B, together with fixed random seeds for PyTorch and NumPy.

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

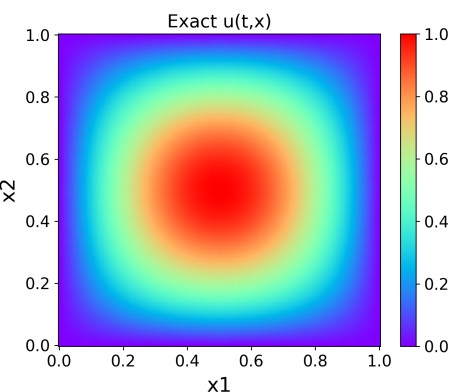 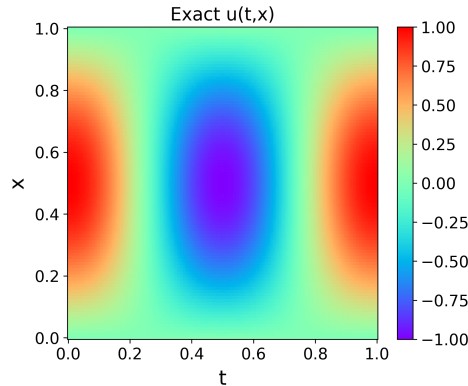

Figure 6: **Ground truths for 5d Poisson and Wave examples.** The left figure is the ground truth of 5d Poisson slice at $(x_3, x_4, x_5) = (0.5, 0.5, 0.5)$. The right one is the ground truth of 1d wave equation when $c = 2.0$.

## A   DETAILS OF PDE EXAMPLES

In this appendix, we detail the PDE setups used in the main paper: Poisson, Wave, Viscous Burgers, and Linear Transport.

### A.1   POISSON EQUATION

We consider the Poisson problem with homogeneous Dirichlet boundary conditions:

$$
\begin{cases}
-\Delta u(\mathbf{x}) = f(\mathbf{x}) & \mathbf{x} \in \Omega, \\
u(\mathbf{x}) = 0 & \mathbf{x} \in \partial\Omega.
\end{cases}
\tag{7}
$$

where $\Omega = [0,1]^d$ and $\mathbf{x} = (x_1, \ldots, x_d)$. We use the manufactured solution

$$
u(\boldsymbol{x}) = \prod_{i=1}^{d} \sin(\pi x_i),
\tag{8}
$$

for which

$$
-\Delta u = d\pi^2 \prod_{i=1}^{d} \sin(\pi x_i) = f(\boldsymbol{x}).
\tag{9}
$$

In the main experiments, we test 5d Poisson and 10d Poisson. Figure 6 shows a 2D slice of 5d Poisson $u$ with respect to $(x_1, x_2)$ while fixing $(x_3, x_4, x_5) = (0.5, 0.5, 0.5)$.

### A.2   WAVE EQUATION

Wave equation is a time-dependent PDE that takes the form:

$$
\begin{cases}
u_{tt}(t, \mathbf{x}) = c^2 \Delta u & \mathbf{x} \in (0,1)^d, t \in [0,1] \\
u(0,t) = u(1,t) = 0 & t \in [0,1] \\
u(\mathbf{x}, 0) = \prod_{i=1}^{d} \sin(\pi x_i), u_t(\mathbf{x}, 0) = 0 & \mathbf{x} \in [0,1]^d
\end{cases}
\tag{10}
$$

where $c$ is the wave speed. In our experiments, we test 1d with $c = 2.0, 5.0, 10.0$ and 2d with $c = 2.0$. The analytical form of Wave equation is $u(t, \mathbf{x}) = \prod_{i=1}^{d} \sin(\pi x_i) \cos(\sqrt{d}\pi c t)$. Figure 6 shows the ground truth figure of 1d Wave equation when $c = 2.0$.

### A.3 VISCOUS BURGERS

The Burgers equation is a fundamental nonlinear PDE combining advection and diffusion, used as a prototype for shock formation and turbulence modeling. We consider the following Viscous Burgers:

$$\begin{cases} u_t + uu_x = \nu u_{xx} & x \in [-1,1], t > 0 \\ u(-1,t) = 0, u(1,t) = 0 & t \geq 0 \\ u(x,0) = -\sin(\pi x) & x \in [-1,1] \end{cases} \tag{11}$$

where viscosity $\nu = \frac{0.01}{\pi}$. With such small viscosity, the solution behaves almost inviscid: gradients steepen rapidly and form very thin viscous layers (shock transitions). Similarly, we set $T = 1$ and $t \in [0,1]$. Analytical solution is introduced in (Basdevant et al., 1986). Gound truth figure is shown in main text, see Figure 4.

### A.4 LINEAR TRANSPORT

Linear Transport (advection—equation) describes a profile being carried along characteristics at velocity without changing shape. The 1d example we use in the main paper takes form as:

$$\begin{cases} u_t + cu_x = 0 & x \in \Omega, t > 0 \\ u(0,t) = u(4,t) = 0 & t \geq 0 \\ u(x,0) = u_0(x) & x \in \Omega \end{cases} \tag{12}$$

where we consider $c = 10$, $\Omega = [0,4]$, $T = 1, t \in [0,1]$ and initial condition $u_0(x)$ as:

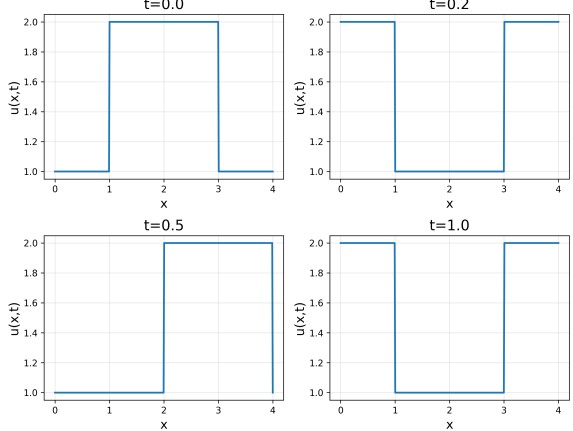

Figure 7: 1d Linear Transport solution profiles when $t = 0, 0.2, 0.5, 1.0s$.

$$u_0(x) = \begin{cases} 2, & 1 \leq x < 3, \\ 1, & \text{otherwise.} \end{cases}$$

The piecewise constant profile induces discontinuities. Given the initial condition, the analytical solution is $u = u_0((x - ct) \mod 4)$. That is, the initial profile simply translates to the right at constant speed $c$ without deformation. We show the solution profiles with respect to $t = 0, 0.2, 0.5, 1.0s$ in Figure 7 We also test another form of 1d Linear Transport, see details in Section C.

## B TRAINING DETAILS

**Data and seed.** For each PDE, we randomly sample according to Gaussian distribution $N_f$ collocation points in the interior domain and $N_b$ points on the boundary. For time-dependent PDEs, we additionally sample $N_{ic}$ points from the initial condition. Values of $N_f, N_b$ and $N_{ic}$ have been shown in main text. For Poisson and Linear Transport problems, we normalize data into $[-1, 1]$ before sending into the model.

$$\tilde{x} = 2\left(\frac{x - x_{\min}}{x_{\max} - x_{\min}}\right) - 1$$

We ensured reproducibility by fixing the random seeds of both NumPy and PyTorch. In particular, `np.random.seed(1234)` and `torch.manual_seed(1234)` were used to control ran-

domness in CPU and single-GPU computations. This setup guarantees that data sampling, weight initialization, and training outcomes remain consistent across repeated runs.

**Router outputs.**  For dense MoE structure, the router outputs mixture weights via a softmax (see equantion 13) with temperature $\tau > 0$. As $\tau \to 0^+$, the distribution becomes more peaked (approaching a one-hot assignment), while $\tau \to \infty$ yields a uniform distribution. In our experiments, for Poisson and Viscous Burgers, we set $\tau = 1.0$ while for Wave and Transport, we set $\tau = 0.5$.

$$\mathrm{softmax}_\tau(z_i) \;=\; \frac{\exp\!\left(\frac{z_i}{\tau}\right)}{\sum_{j=1}^{K} \exp\!\left(\frac{z_j}{\tau}\right)}, \quad i = 1, \ldots, K \tag{13}$$

where $z_i$ is the original output of the router and $K$ is the number of experts.

**Loss function.**  We consider a generic time-dependent PDE written implicitly as

$$\begin{cases} \mathcal{F}\big(x, t, u, \nabla u, \nabla^2 u\big) = 0, & (x, t) \in \Omega \times (0, T], \\ \mathcal{B}[u](x, t) = 0, & (x, t) \in \partial\Omega \times (0, T], \\ u(x, 0) = u_0(x), & x \in \Omega, \end{cases} \tag{14}$$

where $\mathcal{F}$ encodes the governing PDE, $\mathcal{B}$ specifies the boundary condition (Dirichlet/Neumann/periodic), and $u_0$ is the initial condition.

A PINN $u_\theta$ minimizes the composite loss

$$\mathcal{L}(\theta) = w_f \, \mathcal{L}_{\mathrm{PDE}} + w_{bc} \, \mathcal{L}_{\mathrm{BC}} + w_{ic} \, \mathcal{L}_{\mathrm{IC}}, \tag{15}$$

Here $w_f, w_{bc}, w_{ic} > 0$ are scalar weights that balance the PDE residual, boundary, and initial terms, controlling the trade-off among them. The loss is obtained with collocation points $\{(x_f^{(i)}, t_f^{(i)})\}_{i=1}^{N_f}$, boundary points $\{(x_{bc}^{(j)}, t_{bc}^{(j)})\}_{j=1}^{N_{bc}}$, and initial points $\{x_{ic}^{(k)}\}_{k=1}^{N_{ic}}$:

$$\mathcal{L}_{\mathrm{PDE}} = \frac{1}{N_f} \sum_{i=1}^{N_f} \left( \mathcal{F}\big(x, t, u_\theta, \nabla u_\theta, \nabla^2 u_\theta\big) \right)^2 \Big|_{(x_f^{(i)}, t_f^{(i)})}, \tag{16}$$

$$\mathcal{L}_{\mathrm{BC}} = \frac{1}{N_{bc}} \sum_{j=1}^{N_{bc}} \big( \mathcal{B}[u_\theta] \big)^2 \Big|_{(x_{bc}^{(j)}, t_{bc}^{(j)})}, \tag{17}$$

$$\mathcal{L}_{\mathrm{IC}} = \frac{1}{N_{ic}} \sum_{k=1}^{N_{ic}} \big( u_\theta(x_{ic}^{(k)}, 0) - u_0(x_{ic}^{(k)}) \big)^2. \tag{18}$$

All derivatives are obtained via automatic differentiation.

**Two-stage optimization.**  We adopt a two-stage scheme: Adam warm-up followed by L–BFGS refinement. We first optimize the network parameters with Adam ($\mathrm{lr} = 10^{-6}, 5 \times 10^{-4}, 10^{-4}, 10^{-3}$ for Viscous Burger, Poisson, Transport and Wave respectively), updating at each training step:

$$\mathcal{L} \;=\; w_f \, \mathcal{L}_{\mathrm{PDE}} \;+\; w_{bc} \, \mathcal{L}_{\mathrm{BC}} \;+\; w_{ic} \, \mathcal{L}_{\mathrm{IC}}.$$

In our implementation, $w_{bc}$ and $w_{ic}$ is fixed during this phase but $w_f$ is dependent on experiments. For Viscous Burgers, we set $w_{ic} = 10.0$ wile fix the rest weights to $1.0$. For Poisson, $w_{bc} = 5000.0$ and fix $w_f = 1.0$. For Wave, we fix $w_f = 1.0$ while fix others equal to $100.0$. Lastly, for Linear Transport, we fix $w_{ic} = 100.0, w_{bc} = 10.0$. We linearly anneal the PDE residual weight from $w_f^{\mathrm{init}} = 0.01$ to $w_f^{\mathrm{final}} = 1.0$.

$$w_f(e) \;=\; w_f^{\mathrm{init}} \;+\; \big( w_f^{\mathrm{final}} - w_f^{\mathrm{init}} \big) \min\!\left( \frac{e}{T_{\mathrm{anneal}}}, 1 \right),$$

where $e$ is the current Adam step and $T_{\mathrm{anneal}} = 0.75 \, n_{\mathrm{Adam}}$. Thus $w_f$ increases linearly from $0.01$ at $e = 0$ to $1.0$ at $e \geq T_{\mathrm{anneal}}$, after which it remains at $1.0$.

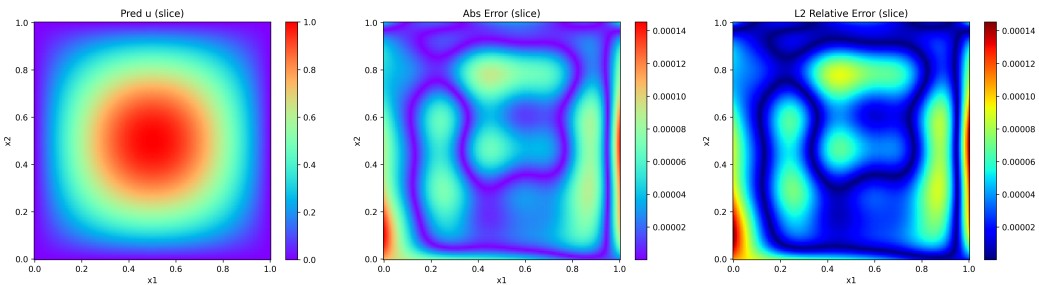

Figure 8: Predicted solutions and error plots for 5d Poisson with single expert module and $r = 4$.

Table 3: $\ell_2$ errors of 5d Poisson with different $r = 1, 2, 3, 4, 5$.

| **Type** | $r = 1$ | $r = 2$ | $r = 3$ | $r = 4$ | $r = 5$ |
|---|---|---|---|---|---|
| 5d Poisson | $7.1881 \times 10^{-4}$ | $2.6559 \times 10^{-4}$ | $1.8219 \times 10^{-4}$ | $1.5252 \times 10^{-4}$ | $3.1061 \times 10^{-4}$ |

We employ cosine annealing for the learning rate with `CosineAnnealingLR` (`T_max` $= 20{,}000$, `eta_min` $= 10^{-6}$), calling the scheduler at every step. Training steps $n_{\text{Adam}}$ varies as experiments. For Viscous Burgers, Wave and Poisson, we set $n_{\text{Adam}} = 10{,}000$ while for Linear Transport, we set $n_{\text{Adam}} = 15{,}000$.

After the Adam warm-up, we switch to `torch.optim.LBFGS` with settings: `max_iter` $= 20{,}000$, `tolerance_grad` $= 10^{-9}$, `tolerance_change` $= 10^{-12}$, `history_size` $= 100$, and strong-Wolfe line search (`line_search_fn` = "strong_wolfe"). Following standard practice, we define a closure that recomputes the loss and its gradients; the PDE and boundary point sets are *fixed once* at the start of this phase (20,000 interior collocation points, 5,000 boundary points and 5,000 initial points) except for Viscous Burgers example where same sampling points are used as Adam. We use the same loss weighting as in Adam.

Unlike the Adam stage (which runs for a fixed number of steps), the L–BFGS stage proceeds until the optimizer's internal convergence criteria are met or `max_iter` is reached. Concretely, L–BFGS terminates early when the gradient norm falls below `tolerance_grad` ($10^{-9}$) or when the change in the objective is smaller than `tolerance_change` ($10^{-12}$), as determined by the strong-Wolfe line search and quasi-Newton updates. Therefore, the number of effective L–BFGS steps is not fixed across runs or PDEs.

## C  ADDITIONAL RESULTS AND EXPERIMENTS

### C.1  EXTENDED RESULTS FOR MAIN EXPERIMENTS

**Poisson and Wave.**  $\ell_2$ relative errors about Poisson experiments are shown here. The Figure 8 demonstrates predicted solution, absolute error and $\ell_2$ relative error about 5d Poisson with single expert module and $r = 4$. The $\ell_2$ errors of 5d Poisson with different $r = 1, 2, 3, 4, 5$ are recorded in Table 3. For 10d Poisson, $\ell_2$ relative error achieve $10^{-3}$ on average. Even when $r = 1$, it obtains $1.0487 \times 10^{-2}$. For Wave equation, Figure 9 shows predicted solutions, absolute errors and relative $\ell_2$ error plots for 1d Wave when $c = 2.0$. Given this setting, $\ell_2$ relative error achieves $2.3779 \times 10^{-4}$. For 2d Wave, an error of $2.4697 \times 10^{-2}$ can be obtained.

**Viscous Burgers.**  We first present domain decomposition results for $K = 4$ and $K = 5$. As shown in Figure 11 and Figure 12, introducing additional experts brings only limited new information to the decomposition. However, the experts attempt to further partition the small triangular area when $t \in [0, 0.3]$, with Expert 5 in the $K = 5$ case showing the most evident specialization. Overall, 3D achieves an $\ell_2$ relative error of approximately $4.33 \times 10^{-4}$, which remains nearly unchanged across different numbers of experts.

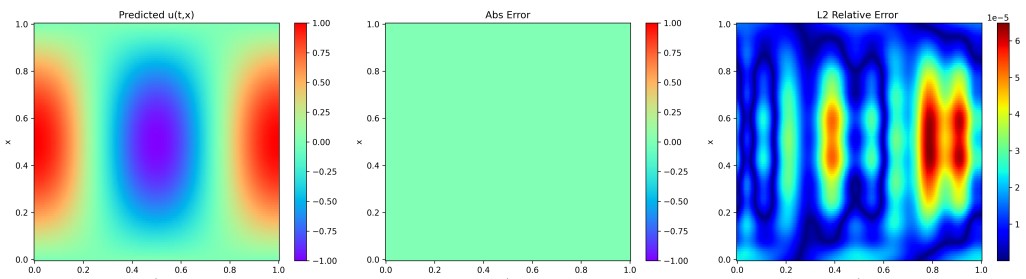

Figure 9: Predicted solutions and error plots for 1d Wave when $c = 2.0$

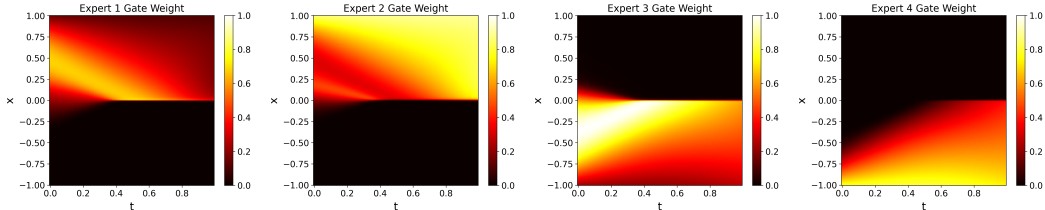

Figure 11: Domain decomposition of Viscous Burgers by $K = 4$.

Then we provide abalation analysis of how $r$ affects the $\ell_2$ relative error. Figure 10 shows the $\ell_2$ error change as training steps up to 15,000 steps for $r = [1, 4, 8, 16]$. Due to the inconsistent of the total training steps, we truncate at the smallest step. For $r = [1, 4, 8, 16]$, the total traning steps are 16500, 15200, 15000, 16800 and the final $\ell_2$ errors are $8.5854 \times 10^{-3}, 1.3682 \times 10^{-3}, 3.3278 \times 10^{-3}, 1.0079 \times 10^{-3}$ respectively. This experiment demonstrates that generally $\ell_2$ error decreases as $r$ increases. However, when it increases to a certain value, its impact on $\ell_2$ error is not that obvious.

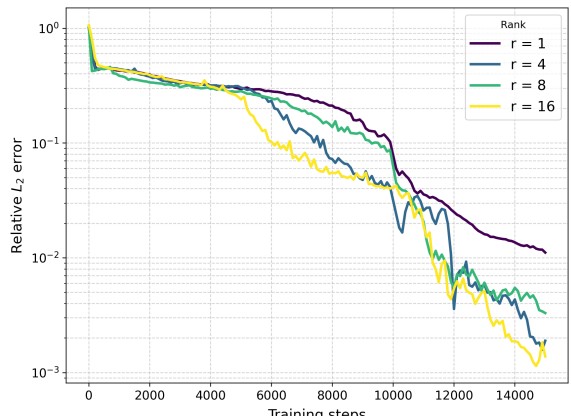

Figure 10: $\ell_2$ **error change as training steps up to** 15,000 **steps for** $r = [1, 4, 8, 16]$**.** For $r = [1, 4, 8, 16]$, the total traning steps are 16500, 15200, 15000, 16800 and the final $\ell_2$ errors are $8.5854 \times 10^{-3}, 1.3682 \times 10^{-3}, 3.3278 \times 10-3, 1.0079 \times 10^{-3}$ respectively.

**Linear Transport.** Here we present additional domain decomposition results of case in main text for $K = 2$ and $K = 5$. For $K = 2$ (Figure 15), the solution is roughly split into two subdomains, but compared with the clearer partition when $K = 3$ (Figure 5), the separation is less distinct. When $K = 5$ (Figure 16), a new subdomain emerges, but the fact that Expert 2 consistently receives zero weight indicates that setting $K = 5$ is redundant. Thus, for this example, $K_{optimal} = 4$.

Figure 13 and Figure 14 summarize the consistency and robustness experiments for Burgers and Transport equations. The first two rows of Figure 13 show the Burgers domain decomposition obtained with 2 experts under two representative seeds (2 and 2025), and both clearly align with the shock at $x = 0$ across seeds. Similarly, the first row of Figure 14 shows the Transport results with 3 experts for seed 2025, where the diagonal stripe patterns remain well captured. To further assess robustness, we inject 5% relative Gaussian noise into both the initial and boundary conditions. The bottom row of Figure 13 and the second row of Figure 14 show the corresponding noisy cases for Burgers and Transport, respectively. In both PDEs, the domain decomposition patterns remain

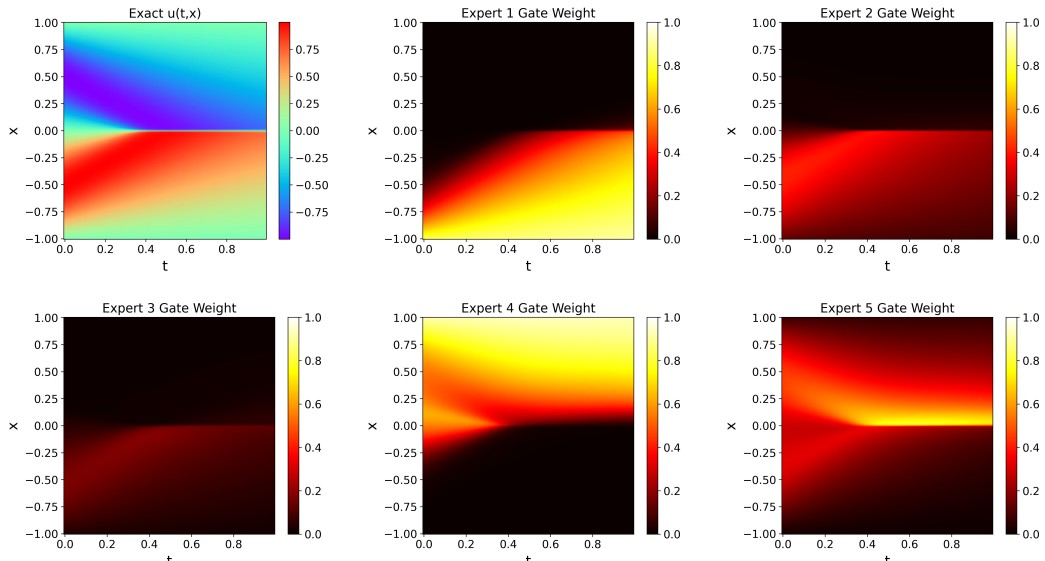

Figure 12: **Domain decomposition of Viscous Burgers by** $K = 5$**.** The top left one is the ground truth figure. The remaining five are domain decomposition for each expert.

stable and nearly identical to the noiseless counterparts, demonstrating strong robustness to data perturbations.

### C.2 NEW EXPERIMENTS

**5d Complex Poisson.** The Poisson example tested in main text is simply the production, we trained the following 5d case with complexity:

$$\begin{cases} -\Delta u(\mathbf{x}) = f(\mathbf{x}) & \mathbf{x} \in \Omega, \\ u(\mathbf{x}) = 0 & \mathbf{x} \in \partial\Omega. \end{cases} \tag{19}$$

where $\Omega = [0,1]^5$ and $\mathbf{x} = (x_1, \dots, x_5)$. We use the manufactured solution

$$u(\boldsymbol{x}) = \sum_{i=1}^{5} \sin(\frac{\pi}{2} x_i)$$

for which

$$-\Delta u = \frac{\pi^2}{4} \sum_{i=1}^{5} \sin(\frac{\pi}{2} x_i) = f(\boldsymbol{x}). \tag{20}$$

Same as before, we also use single expert module to test $VI$. The exact solution of this case is the sum of dimension compoenents, which is consistent with $r = 5$, according to equation 3. Same as our discovery that when $r = 5$, full interpretability is achieved. And the $\ell_2$ relative error is $5.7608 \times 10^{-4}$ given this setting. Figure 17 shows the ground truth, predicted solution and $\ell_2$ relative error plots for $r = 5$.

**2d Poisson with L-shape domain.** To further show the generality of our framework, we test our method on the 2d Poisson on an L-shaped domain following the settings used in SPINNs (Cho et al., 2023):

$$\begin{cases} -\Delta u(x,y) = 1 & \mathbf{x} \in \Omega, \\ u(x,y) = 0 & \mathbf{x} \in \partial\Omega. \end{cases} \tag{21}$$

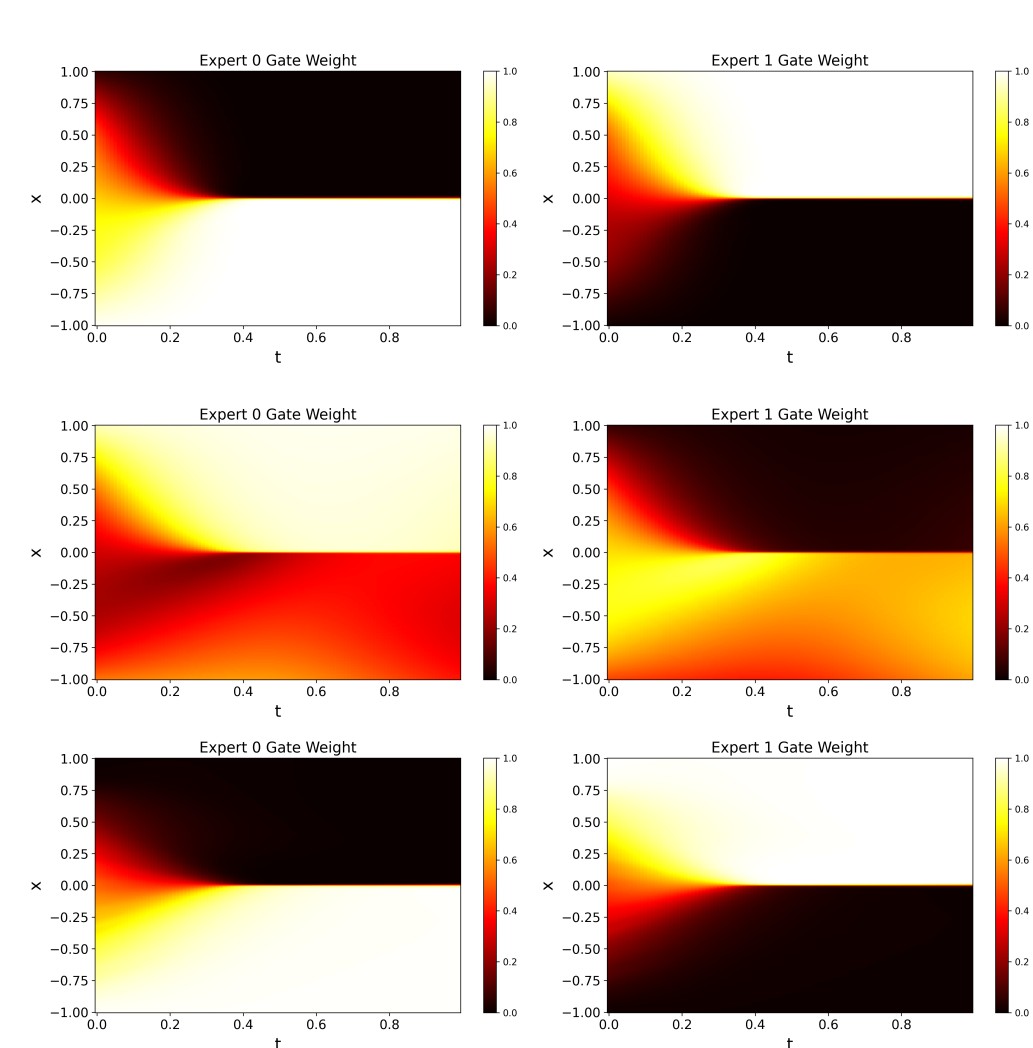

Figure 13: Domain decomposition of the Viscous Burgers equation with 2 experts across different conditions: seed 2 (top), seed 2025 (middle), and 5% noise (bottom).

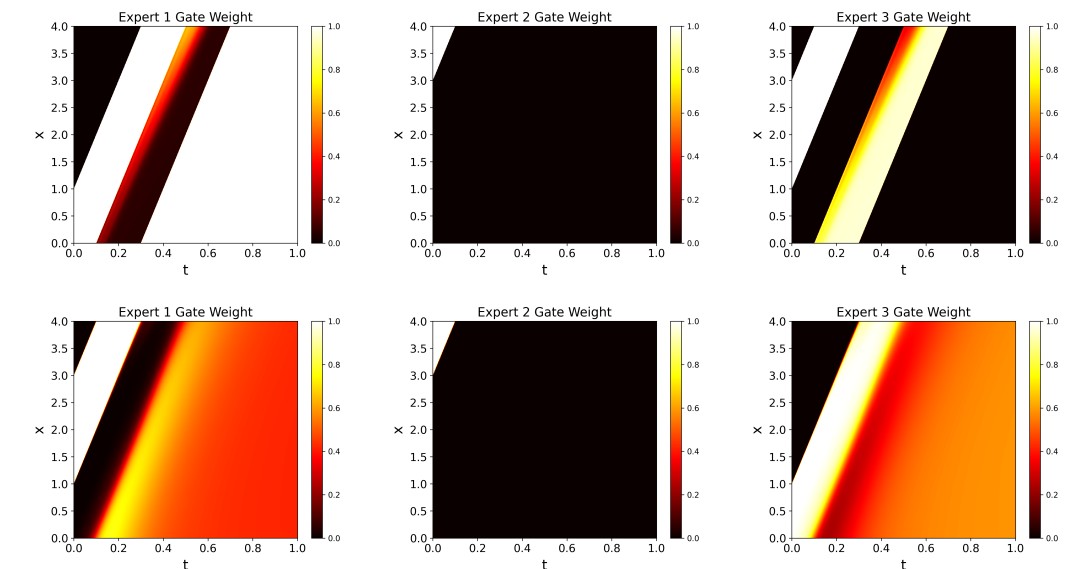

Figure 14: Domain decomposition of the Linear Transport equation with 3 experts across different conditions: seed 2025 (top) and 5% noise (bottom).

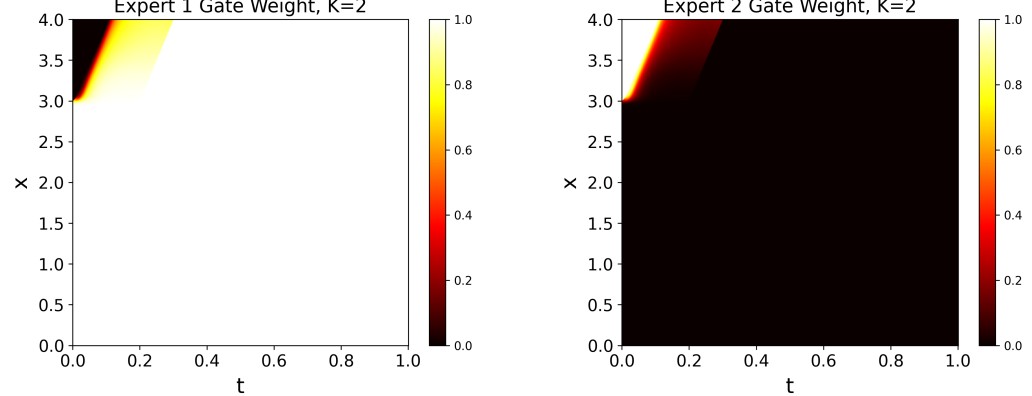

Figure 15: Domain decomposition of Linear Transport with $K = 2$.

where $\Omega = [-1, 1]^2 \setminus [0, 1]^2$. We use high-accuracy finite difference method for solving reference true solutions. Specifically, only $N_{\text{int}} = 10,000$ interior collocation points are drawn from the two disjoint rectangles $[-1, 1] \times [-1, 0]$ and $[-1, 0] \times [0, 1]$, using power-law sampling ($|x| \sim U[0, 1]^{\beta}$, $\beta = 2.5$) to concentrate points near the re-entrant corner. Boundary points are generated uniformly along the polygonal boundary of the L-shape with $N_{\text{bc}} = 200$ points per edge. The model uses a shared MLP with four hidden layers of width $64$ and rank $r = 32$. Under this setting, our model achieves the relative $\ell_2$ error of $2.5520 \times 10^{-2}$ while SPINNs achieves the relative $ell_2$ error of $2.9121 \times 10^{-2}$. The result of 3D is shown in Figure 18

**Fine-tuning Across Dimensions** To evaluate the transferability of our model, we test fine-tuning a 5d model on the 8d Poisson problem. A model is first trained on the 5d Poisson equation using the same settings in Appendix B. The learned parameters are then used to initialize a model for the 8D Poisson problem. For comparison, we also train an 8d Poisson model from scratch under identical settings.

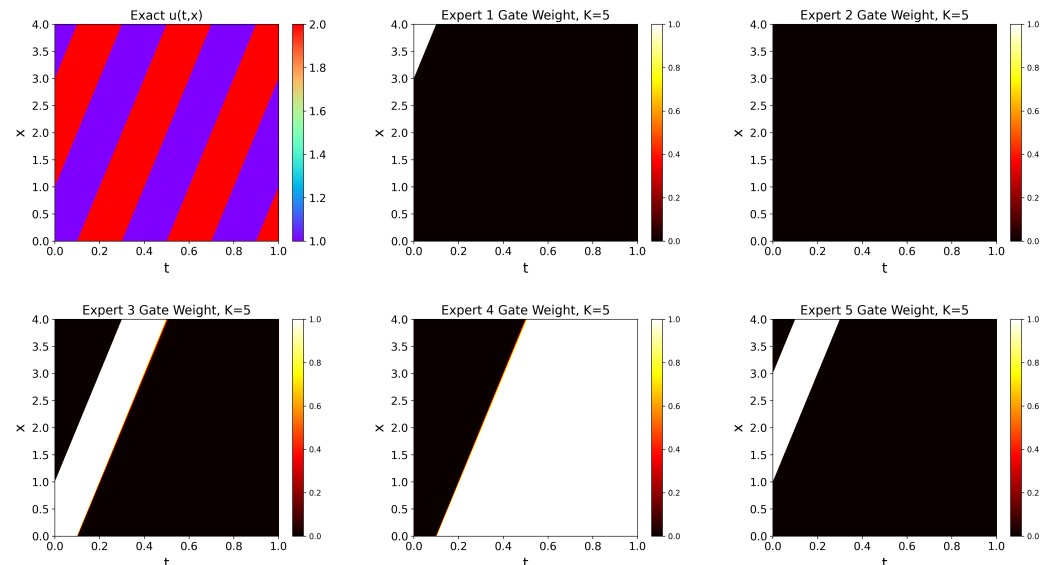

Figure 16: **Domain decomposition of 1d Linear Transport by** $K = 5$. The top left one is the ground truth figure. The remaining five are domain decomposition for each expert.

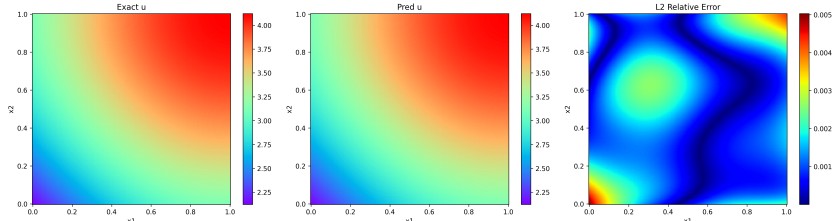

Figure 17: Ground truth, predicted solution and $\ell_2$ relative error plots for 5d Poisson with $r = 5$

Table 4 summarizes the results. Fine-tuning reduces the optimization cost: the fine-tuned 8d model converges in fewer epochs and requires less training time. At the same time, this model achieving a better relative $\ell_2$ error than the model trained from scratch. This confirms that the shared-MLP structure captures reusable low-dimensional patterns that remain meaningful when the dimensionality increases.

**Linear Transport.** We consider another form of 1d Linear Transport with smooth domain. The PDE form is as follow:

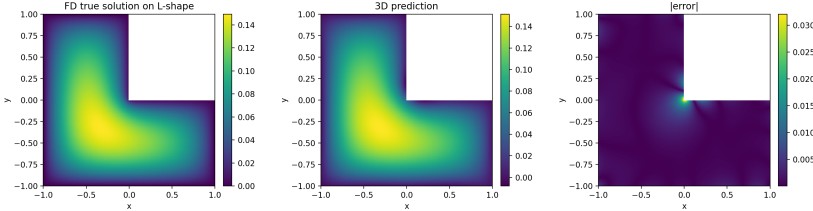

Figure 18: Ground truth, predicted solution and absolute error plots for 2d Poisson with L-shape domain.

Table 4: Fine-tuning from 5d to 8d Poisson. The separable structure allows cross-dimensional transfer, resulting in faster convergence and better accuracy.

| Model | Epochs | Adam Time (s) | Total Time (s) | $\ell_2$ Error ($10^{-4}$) |
|---|---|---|---|---|
| 8d (from scratch) | 11900 | 1486.75 | 1793.56 | 8.9426 |
| 8d (fine-tuned) | 11300 | 1397.22 | 1599.84 | 5.7450 |

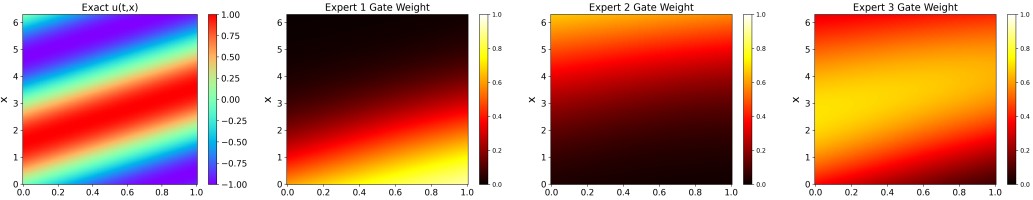

Figure 19: Domain decomposition of 1d Linear Transport with smooth domain by $K = 3$.

$$\begin{cases} u_t + cu_x = 0 & x \in [0, 2\pi], t \in [0, 1] \\ u(0, t) = u(2\pi, t) & t \in [0, 1] \\ u(x, 0) = u_0(x) = \sin(x) & x \in [0, 2\pi] \end{cases} \quad (22)$$

The analytical solution is $u(t, x) = \sin(x - ct) = \sin(x)\cos(ct) - \cos(x)\sin(ct)$. Using single expert module with $r = 5$, we get $L_2$ error $1.5159 \times 10^{-3}$. We obtain error $1.3409 \times 10^{-3}$ when using three experts with $r = 5$. Though with smooth region, 3D partitions the domain into subdomains separated by diagonal stripes similar to that in the ground truth. The Figure 19 shows the domain decomposition results when $K = 3$. In this experiment, we find $K_{optimal} = 3$. We also test $VI$ of this example. When $r = 1, 2, 3, 4, 5$, $VI = 0.8955, 0.8614, 0.9242, 0.9887, 0.9950$, further indicating good interpretability even for small $r$.

## D  USE OF LARGE LANGUAGE MODELS (LLMS)

In preparing this work, we made limited use of Large Language Models (LLMs) as auxiliary tools. Specifically:

- **Editing and Polishing** We used an LLM (ChatGPT) to polish the language of the paper, including improving grammar, readability, and stylistic clarity. The scientific content, arguments, and conclusions were entirely authored by us.

- **Literature search assistance** We used the LLM to help identify relevant references and related work. All final references were cross-checked and selected manually by the authors.

- **Coding assistance** For certain implementation details, we consulted the LLM to generate small code snippets (e.g., plotting utilities, debugging suggestions). The core research code, experimental design, and implementation were created and validated by the authors. The LLM was not involved in the generation of research ideas, methodological design, experimental analysis, or the writing of scientific contributions. Its role was strictly supportive, and final decisions on wording, citations, and code were made by the authors.

