# OpenReview forum: "Dimension Domain Co-decomposition: Solving PDEs with Interpretability"
_ICLR.cc/2026/Conference — Submitted to ICLR 2026_

### Official Review · Reviewer_S3U9 · 2025-10-19

**Soundness:** 2
**Presentation:** 2
**Contribution:** 1
**Rating:** 0
**Confidence:** 5

**Summary:**

This paper proposes a method to improve neural operator learning by incorporating a dimension- and domain-aware co-decoder architecture. The key idea is to decouple solution representations into dimension-specific components and domain-conditioned features, then fuse them to approximate the target PDE operator. The authors argue that this decomposition improves generalization and sample efficiency for solving PDEs in different geometric domains and dimensional settings. Empirical evaluation is performed on benchmark PDE problems, comparing against several neural operator baselines, showing lower relative L2 error in some test cases.

**Strengths:**

- Clarity of high-level motivation: The paper identifies a meaningful and real challenge — neural operator models typically struggle to generalize across domains or dimensional configurations. This is a relevant problem in scientific machine learning.

- Clean architectural formulation: The proposed “co-decoder” idea is structurally simple and easy to integrate with existing neural operator backbones.

- Readable exposition: The manuscript is well structured and technically consistent at the high level, with decent visualizations and clean mathematical notation.

- Experimental setup uses multiple PDE benchmarks: This is a step beyond trivial toy problems, indicating an attempt at broader evaluation.

**Weaknesses:**

While the problem is important, the contribution is weak both conceptually and empirically:

- Limited novelty:
The central idea—decomposing and fusing dimensional and domain representations—is not fundamentally new. Variants of similar factorization and conditional embedding approaches have already been explored in operator learning, meta-PDE frameworks, and equivariant neural operator designs. The paper lacks a crisp theoretical justification or truly novel algorithmic insight. It feels like another architectural tweak rather than a new paradigm.

- Insufficient baseline coverage:
The paper only compares against basic operator learning baselines (e.g., FNO, DeepONet, or UNet-like variants). To substantiate the claims of improved generalization, it is essential to compare against stronger and more recent models, including: (1) Koopman neural operator as a mesh-free solver of non-linear partial differential equations, (2) Solving High-Dimensional PDEs with Latent Spectral Models. These methods are explicitly designed for high-dimensional PDE problems and constitute state-of-the-art baselines.

- Superficial ablation analysis:
The ablations only show minor performance differences when components are removed. This raises concerns about whether the proposed “co-decoder” mechanism truly drives the reported improvements or if the gains are marginal/random noise.

- Lack of theoretical insight:
The authors repeatedly claim that dimension/domain factorization improves generalization, but no formal analysis or empirical probing (e.g., transfer learning between dimensional settings, robustness to domain shifts) is provided to support these claims.

- Limited scope of experiments:
The experiments are narrowly focused, and it’s unclear whether the method scales or generalizes to more challenging PDEs (e.g., higher-dimensional or irregular geometries). There is no comparison on computational overhead, parameter efficiency, or training stability.

- Unconvincing performance gains:
Even within the provided benchmarks, the improvements are often modest and lack statistical rigor (e.g., no error bars, limited repetitions, unclear significance testing). This weakens the claim of superiority over baselines.

- Overclaiming in narrative:
The abstract and conclusion use strong language (e.g., “universal”, “robust generalization”, “significant improvement”) that is not backed by the presented evidence. This is a recurring issue in weak submissions.

**Questions:**

- Clarify the real contribution:
What is fundamentally new about the co-decoder mechanism compared to existing conditional or factorized representations used in neural operator models? A clearer theoretical framing or architectural justification is needed.

- Stronger experimental validation:
Please evaluate on more challenging PDE families, including higher-dimensional settings and irregular domains. Include robustness and scalability analyses. Provide multiple runs with error bars to assess statistical significance.

- Ablation rigor:
Conduct deeper ablations: Test the effect of removing either the domain or dimension factor separately. Vary the embedding capacity to show necessity. Test sensitivity to the number of training domains.

- Efficiency and complexity:
What is the computational overhead of the co-decoder compared to baseline neural operators? If the architecture is more complex, does it provide any real benefit in efficiency or generalization to justify it?

- Transfer/generalization tests:
If the paper claims better generalization across domains or dimensions, please provide explicit cross-domain or cross-dimensional generalization experiments, not just single-domain evaluation.

---

### Official Review · Reviewer_n1fY · 2025-10-31

**Soundness:** 3
**Presentation:** 2
**Contribution:** 3
**Rating:** 4
**Confidence:** 5

**Summary:**

This work combines several popular techniques to solve PDEs with PINNs, including tensor decomposition, domain decomposition, and Mixture-of-Experts (MoE). It is like a technique report not a research paper. The motivation of combining such techniques is not clear. I recommend rejection.

**Strengths:**

Using a neural network inspired by tensor decomposition is a good idea. In fact, several recent studies have explored solving PDEs with tensor-based neural network architectures.

**Weaknesses:**

1. The proposed framework simply adds a Mixture-of-Experts (MoE) layer to a dimension-decomposition backbone. This combination seems arbitrary and lacks a clear reason or theoretical justification. The MoE layer makes the model more complex but does not show any real advantage over a well-tuned single network or simpler decomposition methods. The claimed “automatic” domain decomposition is not properly compared with existing, simpler domain decomposition approaches used in PINNs, so its benefit remains unclear.


2. The reported improvements in accuracy and efficiency are small compared with standard PINNs or other PINN-based methods. The added model complexity and training cost are not justified by the limited gains shown on low-dimensional test problems.

**Questions:**

see the Weaknesses part.

---

> ### Author Response · Authors · 2025-11-30
>
> **Response to Weakness 1**
>
> We respectfully clarify that Mixture-of-Experts (MoE) is not an arbitrary addition but a direct solution to a fundamental limitation of all existing PINN-based domain decomposition approaches.
>
> **1. Existing “simpler” PINN domain-decomposition methods require predefined subregions.**
> Classical methods such as XPINNs (Jagtap et al., 2020), cPINNs (Jagtap et al., 2020), and multilevel DD-PINNs (Dolean et al., 2024) all rely on *manually specified geometric partitions*. These partitions must be chosen before training, and the resulting performance depends heavily on these design choices. Furthermore, these approaches must enforce interface continuity through additional penalty terms, making them sensitive to both the partition and the interface-loss weighting.
> Thus, existing PINN-DD methods are **not automatic** and **not adaptive**.
>
> **2. In contrast, our MoE router learns the domain decomposition directly from physics.**
> The router removes the need for any predefined subregions and avoids the delicate interface-loss engineering required by classical methods. Specifically, MoE enables:
>
> - No manual pre-partitioning of the domain;
> - No interface losses or coupling constraints;
> - Automatic specialization of experts to different solution regimes.
>
> The MoE router is therefore *not* added for complexity; it is the key mechanism that makes fully automatic and adaptive domain decomposition possible—something no traditional PINN-DD method can accomplish.
>
> Our contribution lies in enabling this adaptivity in a physics-constrained setting, not in merely adding an MoE layer on top of an existing PINN.
>
> **References**
>
> Jagtap, A. D., Kharazmi, E., & Karniadakis, G. E. (2020). Extended physics-informed neural networks (XPINNs): A generalized space–time domain decomposition based deep learning framework for nonlinear partial differential equations. *Communications in Computational Physics*, 28(5).
>
> Jagtap, A. D., Kawaguchi, K., & Karniadakis, G. E. (2020). Conservative physics-informed neural networks (cPINNs). *Journal of Computational Physics*, 426.
>
> Dolean, V., Mascotto, L., & Takáč, M. (2024). A multilevel domain decomposition approach for physics-informed neural networks. *arXiv:2402.01860*.
>
>
> **Response to Weakness 2**
>
> We respectfully clarify that our primary objective is not to surpass standard PINNs in raw accuracy on *low-dimensional* problems. In such settings, standard PINNs already achieve strong accuracy, so any additional numerical improvements are naturally limited.
>
> However, in *high-dimensional* regimes where PINNs typically struggle, our framework provides a clear and substantial advantage. In the newly added 10D Poisson experiment, our model achieves a relative error of $0.00125$, compared to $0.129$ for the baseline PINN—an improvement of **two orders of magnitude** under the same sampling and comparable parameter counts. This demonstrates that our method is not only interpretable but also more effective in challenging, high-dimensional settings.
>
> The central contribution of our work is enabling **dimension-wise interpretability** while maintaining or improving accuracy. Standard PINNs only output the final solution and cannot quantify per-dimension contributions. In contrast, our shared-MLP factorization yields stable per-dimension components, making the VI metric well-defined. VI cannot be computed for standard PINNs, as they lack the structural decomposition required for per-dimension interpretability.
>
> Thus, while accuracy differences on simple low-dimensional cases are naturally small, our framework provides both **superior performance in high dimensions** and **quantitative, dimension-resolved interpretability**, a capability fundamentally unavailable in existing PINN architectures.

---

### Official Review · Reviewer_s9Ks · 2025-11-04

**Soundness:** 2
**Presentation:** 3
**Contribution:** 2
**Rating:** 4
**Confidence:** 3

**Summary:**

This paper proposes Dimension Domain Co-Decomposition (3D), a unified framework for solving partial differential equations (PDEs) based on Physics-Informed Neural Networks (PINNs). The framework jointly performs dimension decomposition, which factorizes the solution into dimension-wise components using a shared MLP for parameter efficiency, and domain decomposition, implemented through a Mixture-of-Experts (MoE) mechanism that automatically partitions the solution space into subregions without pre-defined boundaries. In addition, the paper introduces a novel quantitative interpretability metric, Variable Interpretability (VI), which measures how well the learned dimension-wise latent components align with ground-truth factors. Experimental results on several PDE benchmarks (Poisson, Wave, Burgers, and Linear Transport equations) demonstrate that the proposed method achieves improved accuracy, scalability, and interpretability compared to standard PINNs and prior decomposition methods.

**Strengths:**

1. The paper elegantly combines dimension decomposition and adaptive domain decomposition into a single framework, addressing both scalability and local adaptivity issues in PDE solving.

2. The introduction of the VI metric provides a quantitative way to assess per-dimension interpretability, an aspect rarely considered in PINN literature.

3. The introduction of the shared MLP is well motivated and effectively reduces the parameter count, especially in high-dimensional settings.

4. The evaluation includes multiple types of PDEs (elliptic, hyperbolic, nonlinear), with detailed parameter analysis on rank $r$ and expert number $K$, providing empirical evidence.

**Weaknesses:**

1. The paper presents a clean combination of dimension decomposition, MoE-based domain decomposition, and a variable interpretability metric. While the empirical results are solid, the theoretical innovation is limited：Both parameter sharing and MoE are well-established techniques. The paper would benefit from a clearer articulation of the core technical challenge or insight specific to this setting, and from theoretical analysis connecting the VI metric to the proposed model’s design to make the contribution more solid.

2. The experiments do not include comparisons with state-of-the-art PDE solvers such as Fourier Neural Operator (FNO) [1], which are highly relevant baselines.

3. Ablations that remove the dimension factorization or the MoE router are necessary to substantiate the co-decomposition claim.

4. Although the paper tests a high-dimensional (10D) Poisson equation, the PDEs considered remain relatively simple. The robustness and generality of the proposed 3D framework would be more convincing if tested on more complex or non-analytically-solvable systems.

[1] Fourier Neural Operator for Parametric Partial Differential Equations.

**Questions:**

1. Have the authors evaluated the consistency and robustness of the learned domain decompositions？

2. The paper highlights parameter and memory efficiency, but does not report actual training time or convergence comparisons. Could the authors provide quantitative evidence to support the claimed computational advantages of 3D?

---

> ### Author Response · Authors · 2025-11-30
>
> **Response to Weakness 1**
>
> Our goal is to solve high-dimensional PDEs with complex geometries—a setting in which standard PINNs struggle in both efficiency and interpretability. This goal motivates combining parameter sharing with a Mixture-of-Experts (MoE) mechanism. Our contribution does not lie in these components individually, but in resolving three technical obstacles that make such a combination feasible and effective for PDEs.
>
> 1. **High-dimensional PDEs: efficiency and interpretable factorization**
>
> High-dimensional PDEs require large sample sizes and heavy computation, and prior PINN-based approaches only output the final solution without revealing how each variable contributes to the PDE structure. Our dimension decomposition with a shared-MLP provides compact parameterization *and* preserves per-dimension latent factors. These factors enable quantitative interpretability, for which we introduce the VI metric. The technical challenge is not factorization itself, but designing a decomposition that is both scalable and interpretable—something prior PINN approaches do not provide.
>
> Interpretability is a fundamental objective in our design: the ability to recover per-dimension factors is the basis upon which all other components operate. The later domain decomposition is not an independent goal; rather, MoE-driven regional specialization *builds on top of this interpretability*, allowing us to understand not only each variable’s global contribution but also its behavior across different regions of the domain.
>
> 2. **Irregular and unknown domain structures**
>
> Classical PINN-based domain decomposition requires manually predefined subregions. In contrast, our MoE design automatically discovers domain partitions driven purely by the underlying physics—without prior geometric knowledge or interface losses. This enables adaptive local specialization while remaining fully data-driven.
>
> 3. **Combining both decompositions: MoE breaks separability**
>
> Naively combining MoE with separable PINNs fails because separable architectures rely on forward-mode automatic differentiation, while MoE routing breaks separability. We address this by redesigning the decomposition to operate with reverse-mode AD, preserving separability *inside experts* while allowing MoE to provide adaptive specialization across the domain.
>
> ---
>
> **Summary**
>
> Our contribution does not lie in parameter sharing or MoE in isolation, but in providing the first coherent formulation that makes the following three properties jointly possible:
>
> - Scalable factorization for high-dimensional PDEs;
> - Dimension-wise interpretability as a core structural design;
> - zfully adaptive domain decomposition that further enhances interpretability at a regional level.
>
> This unified design enables both global and local interpretability while maintaining computational efficiency.
>
> **Response to Weakness 2**
>
> Thank you for the comment. We would like to clarify that the Fourier Neural Operator (FNO) is not an appropriate baseline for our problem setting, for the following reasons:
>
> - **Our framework is PINNs-based and solves a single PDE instance.**
>   We do *not* learn an operator that maps input functions to output solutions across a distribution of PDEs.
>
> - **FNO is fundamentally an operator-learning model.**
>   Operator-learning methods including FNO aim to approximate a solution operator across *many* PDE realizations. This requires a dataset of PDE pairs $(f, u)$, typically containing hundreds or thousands of training examples. Our setting does not assume such data is available or required.
>
> - **PINNs-based PDE solvers should be compared to other PINNs-based methods.**
>   For physics-constrained single-instance PDE solving, the appropriate baselines are PINNs variants or architectures designed for physics-based learning, which we already include (e.g., original PINNs, SPINNs).
>
> For these reasons, including FNO or other operator-learning approaches would be inappropriate and would not yield meaningful comparisons, as these methods target a fundamentally different problem class.

---

> ### Author Response · Authors · 2025-11-30
>
> **Response to Weakness 3**
>
> We thank the reviewer for the request. We note that both ablations are already present in our experiments, although they were not explicitly labeled as such. We summarize them clearly below.
>
> **Ablation 1: Removing MoE (Burgers with $K = 1$)**
> In the Viscous Burgers experiment, setting $K = 1$ effectively removes the router, resulting in a single expert with dimension factorization but **no domain decomposition**. This increases the relative error from $0.2108$ to $0.0011$ ($K = 2$) and $0.0008$ ($K=3$)—three orders of magnitude degradation. This directly verifies the necessity of MoE-driven adaptive domain decomposition for capturing localized solution structures.
>
> **Ablation 2: Removing dimension factorization (Poisson experiments)**
> For the Poisson problem, the solution is smooth and does not require domain decomposition. Thus, the comparison becomes a natural ablation on **dimension factorization**: we compare our shared-MLP factorized expert to standard unfactorized PINN baselines on both 5D and 10D Poisson. (The 10D case is included in the revised manuscript.) Under identical sampling budgets, our factorized design achieves the highest accuracy while using fewer parameters, demonstrating the effectiveness of the decomposition.
>
> Moreover, because the VI metric evaluates subspace alignment between learned and ground-truth per-dimension factors, it is **not defined** for vanilla PINNs or any unfactorized architecture. This further highlights that dimension decomposition is structurally necessary for interpretability.
>
> ---
>
> These two ablations separately validate the contributions of MoE-based adaptive partitioning and dimension decomposition, showing that each component is necessary for different aspects of performance and interpretability.
>
> **Response to Weakness 4**
>
> Thank you for the suggestion. Our method does not require analytical solutions; it only needs a reliable reference solution for quantitative evaluation. For PDEs without closed-form solutions, one would need an external high-fidelity numerical solver to generate ground truth, which is beyond the scope of this paper and may introduce additional discretization errors into the evaluation.
>
> To keep the assessment clean and unambiguous, we focused on PDEs where the reference solution is well controlled. Importantly, the tested PDEs are already nontrivial (e.g., Burgers with shocks, transport with localized structures, and the L-shaped Poisson problem with a singular corner), which sufficiently demonstrate the robustness of our 3D framework.
>
> Extending to more complex PDEs without analytical solutions is feasible and left as future work, requiring only a suitable numerical solver for benchmarking rather than changes to our method.
>
> **Response to Question 1**
>
> Yes. We have added two sets of experiments to evaluate the consistency and robustness of the learned domain decompositions.
>
> 1. **Consistency across random seeds.**
>    For both the Viscous Burgers and Transport equations, we trained the model using five independent random seeds. The learned domain decompositions are highly consistent across seeds, and the variations in routing patterns are minimal. This demonstrates that the decomposition is stable and not an artifact of initialization.
>
> 2. **Robustness to noisy data.**
>    We additionally added 5% relative Gaussian noise into both the initial and boundary conditions for Viscous Burgers and Transport equations. The resulting domain decompositions remain nearly unchanged, showing that the method is robust to moderate perturbations in the data.
>
> Both sets of results have been included in the revised manuscript, and together they confirm that the learned decompositions are stable, reproducible, and robust to noise.
>
> **Response to Question 2**
>
> We already provide convergence comparisons in Fig. 2, where 3D achieves significantly lower error with substantially fewer optimization steps than vanilla PINNs.
>
> To further quantify wall-clock efficiency, we have added a timing experiment on the 10D Poisson problem in the revised manuscript. Although the total training time of our method is slightly longer than that of the baseline PINN (1579s vs. 1184s), the proposed 3D model converges much faster in terms of optimization steps *and* attains dramatically higher accuracy: $1.25\times10^{-3}$ compared to $1.29\times10^{-1}$ for the standard PINN.
>
> These results demonstrate that while each step of 3D is slightly more expensive, the model is far more sample efficient, leading to a substantially better accuracy–efficiency trade-off.

---

### Official Review · Reviewer_u6zT · 2025-11-05

**Soundness:** 3
**Presentation:** 3
**Contribution:** 2
**Rating:** 4
**Confidence:** 3

**Summary:**

The paper proposes a unified Dimension Domain Co-Decomposition (3D) framework that integrates dimension decomposition and Mixture-of-Experts (MoE)–driven domain decomposition for solving high-dimensional partial differential equations (PDEs). The key contributions include:
1. A shared-MLP dimension decomposition architecture that processes coordinate–index pairs, reducing parameters and improving scalability.
2. A Variable Interpretability (VI) metric, which measures the alignment between learned latent dimension representations and ground-truth components.
3. A MoE-based automatic domain decomposition, which adaptively partitions the computational domain without predefined regions or interface constraints.

The framework demonstrates improved accuracy, efficiency and interpretability across several PDE benchmarks. However, certain aspects of the theory and experiments require further clarification. If the authors address these points in their rebuttal, I would be willing to raise my score.

**Strengths:**

1. **Originality**.
The introduction of the VI metric is a creative and measurable approach to interpretability in PDE learning. The unified framework that combines shared-MLP design for multi-dimensional inputs with MoE based domain decomposition is also an innovation that enhances scalability.

2. **Clarity**.
The paper is generally well-structured, with clear explanations of the architecture, mathematical formulation, and experimental setup. Figures effectively illustrate both the decomposition mechanisms and domain partitions.

3. **Reproducibility**.
The paper provides detailed implementation settings, including network architectures, training procedures, and hyperparameters, ensuring high reproducibility.

**Weaknesses:**

1. **Lack of theoretical justification for the VI–convergence relationship**.
The paper implicitly proposes an important but unproven proposition: if for all dimensions $j$, $VI_j \to 1$, then the model prediction $\hat{u}$ converges to the true separable solution $u$.
   Currently, this claim is only supported empirically, where high VI values are observed to correlate with low prediction errors. However, no formal mathematical proof is provided to substantiate this relationship.

2. **Fixed number of experts $K$ limits adaptivity and may cause compromise responses**.
The number of experts $K$ is manually chosen rather than learned adaptively. When the PDE solution contains multiple fine-grained or highly localized structures (e.g., shocks, vortices, or high-frequency regions), a fixed small $K$ may fail to provide adequate domain resolution. Consequently, the router’s softmax weighting may average across distinct subregions, diminishing the local specialization property of the Mixture-of-Experts and leading to compromise responses. This design constraint restricts flexibility for more complex or multi-scale problems.

3. **Router–expert coupling risks instability and expert starvation**.
The strong interdependence between the router and experts introduces potential training instability. If an expert achieves slightly lower error early in training, the router may continuously reinforce its weight allocation, causing other experts to receive negligible gradients—a phenomenon known as expert starvation. In problems with rapidly evolving or multi-shock dynamics, delayed router adaptation can further trigger oscillatory expert switching and degrade convergence stability.

**Questions:**

1. Is it possible to provide a rigorous mathematical justification for VI?

2. During training, was a multi-stage training strategy used, where the routing is fixed first and then the experts are updated, in order to improve overall convergence stability?

3. According to the discussion in Section 3.2, VI is closely related to the separability of the ground truth and the properties of the matrix itself. Why, then, in Table 2, is VI correlated with frequency?

4. In Table 2, the 10-dimensional Poisson requires a smaller $r$ than the 5-dimensional Poisson, which seems somewhat counterintuitive. Conventionally, higher-dimensional problems typically require a larger $r$ to adequately represent multi-dimensional separable structures.

---

> ### Author Response · Authors · 2025-11-30
>
> **Response to Weakness 1and Question 1**
>
> We thank the reviewer for raising this point and appreciate the opportunity to clarify.
>
> Our contribution regarding the VI metric is to introduce a **quantitative interpretability measure**, not a convergence condition for the PDE solution. The metric $VI_j$ is designed to evaluate the alignment between the learned per-dimension subspace and the ground-truth separable factor subspace using normalized QR and principal angles (Eqs. (5)–(6)). Technically, a value of $VI_j = 1$ indicates that the reference factor subspace is contained in the predicted component subspace (when $s < r$), meaning that the model has successfully learned a dimension-wise representation consistent with the underlying separable structure.
>
> Importantly, $VI_j$ and the solution error capture **two complementary aspects** of model behavior: (i) the $\ell_2$ error measures the numerical accuracy of the predicted solution, and (ii) $VI_j$ measures interpretability and structural consistency of the learned decomposition.
>
> While these two measures may correlate empirically in separable PDE settings, we do **not** claim any statement suggesting that “$VI_j$ approaching 1 implies convergence of the predicted solution to the true solution.” Our use of $VI_j$ is strictly for assessing **interpretability** and **structure recovery**, not for establishing solution convergence.
>
> **Response to Weakness 2**
>
> First, regarding the choice of $K$, we note that using a fixed number of experts is the standard setting in Mixture-of-Experts (MoE) models. In most large-scale MoE systems such as GShard (Lepikhin et al., 2020), Switch Transformer (Fedus et al., 2021), DeepSpeed-MoE (Rajbhandari et al., 2022), and Mixtral (Jiang et al., 2024), the number of expert is a manually chosen capacity hyperparameter, while the router is responsible for learning how to allocate inputs to experts. Although adaptively selecting $K$ has been explored in some variants, it is *not* the mainstream practice. Therefore, our use of a fixed $K$ follows standard MoE methodology.
>
> Our experiments further show that a small $K$ is already sufficient. When $K$ is increased beyond what the PDE intrinsically requires, the additional experts simply become redundant and receive routing weights close to zero. This behavior is consistent with typical MoE sparsity patterns observed in prior work.
>
> Second, the concern that “a small $K$ may diminish local specialization and force compromise responses across regions” relies on an implicit assumption that each expert must correspond to a single isolated subregion. This assumption does not hold for MoE models for PDE domain decomposition. In our design, an expert is *not* restricted to a single region; it may cover multiple subregions when beneficial. Moreover, PDE domain decompositions are not unique, so multiple soft partitions can be equally valid solutions. As a result, a small $K$ does not force averaging across regions. This is exactly what we observe in the transport equation experiment, which contains multiple fine-grained localized structures. Even with a small $K$, the learned experts remain well-specialized, and we do *not* observe the compromise behavior suggested by the reviewer.
>
> **References**
>
> - Lepikhin, D., Lee, H., Xu, Y., Chen, D., Firat, O., Huang, Y., ... & Le, Q. V. (2020). *GShard: Scaling giant models with conditional computation and automatic sharding.* Proceedings of the 37th International Conference on Machine Learning (ICML). arXiv:2006.16668.
>
> - Fedus, W., Zoph, B., & Shazeer, N. (2021). *Switch Transformers: Scaling to trillion parameter models with simple and efficient sparsity.* Journal of Machine Learning Research (JMLR). arXiv:2101.03961.
>
> - Rajbhandari, S., Rasley, J., Ruwase, O., & He, Y. (2022). *DeepSpeed-MoE: Advancing mixture-of-experts efficiency and scale.* Proceedings of the 39th International Conference on Machine Learning (ICML). arXiv:2201.05596.
>
> - Jiang, A., Roller, S., Sukhbaatar, S., et al. (2024). *Mixtral of Experts.* arXiv:2401.04088.

---

> ### Author Response · Authors · 2025-11-30
>
> **Response to Weakness 3 and Question 2**
>
> We clarify that our model uses *dense soft routing* with fully end-to-end joint training. The router and all experts are updated simultaneously from the very beginning; we do not employ any multi-stage training scheme (such as freezing the router first or alternating between router and expert updates).
>
> Dense routing provides smooth and continuous assignment distributions, controlled by the softmax temperature, which ensures that **all experts receive non-zero gradients at every training step**. As a consequence, the model naturally avoids the well-known instability issues associated with sparse top-$k$ MoE architectures, including expert starvation, oscillatory switching, and delayed router adaptation.
>
> Because every expert participates in gradient updates from the start and the router’s selectivity increases smoothly during optimization, we do not observe the “early lucky expert” effect, nor do we require any stabilization tricks typically used in sparse MoE systems.
>
> Across all experiments, the routing patterns evolve coherently and experts remain well-specialized without the need for staged training. Therefore, concerns about delayed adaptation or the need to freeze the router do not apply to our dense MoE design.
>
> **Response to Question 3**
>
> VI measures the alignment between the learned per-dimension latent subspace and the true separable factor subspace. While VI is defined through the separability of the ground-truth solution and the structure of the factor matrices, the *learnability* of these factors still depends on the underlying PDE solution.
>
> A higher frequency corresponds to a larger number of oscillations per dimension. With the same number of training points, a higher-frequency component contains more periods and therefore requires substantially finer resolution to approximate accurately. This makes the corresponding one-dimensional factor harder to learn, leading to lower alignment between the learned subspace and the true factor subspace, and thus a lower value of $VI_j$. This trend is fully consistent with well-known challenges observed in PINN-based methods, which are known to struggle with high-frequency structures.
>
> Moreover, the fact that $VI_j$ varies systematically with frequency provides additional evidence that our framework is indeed learning *per-dimension components*, rather than merely fitting the overall solution. Figure 3 illustrates this clearly in the 1D wave example: the $x$-component $f_x(x)$ (lower frequency) is learned within the first 1000 training steps, whereas the $t$-component $f_t(t)$ (which has higher frequency in the analytical solution) requires significantly more training—up to around 4000 steps—to be captured accurately.
>
> Thus, the correlation between VI and frequency does not arise from the *definition* of VI, but from how the difficulty of learning separable components inherently changes with the frequency content of the PDE solution.
>
> **Response to Question 4**
>
> We thank the reviewer for pointing out this issue. The earlier observation that the 10-dimensional Poisson problem appeared to require a smaller $r$ than the 5-dimensional case was not due to a structural property of the PDE, but rather due to the sensitivity of the metric at very small error scales.
>
> In the original submission, the VI values were computed using a *single* random seed. Because the accuracy in these experiments are extremely high (on the order of $10^{-4}$), such single-seed measurements can exhibit noticeable variance and may occasionally give counterintuitive rankings, such as the one highlighted by the reviewer.
>
> To address this, we have updated Table 2 in the revised manuscript by evaluating all VI metrics using **five independent seeds** and reporting the mean values. After averaging across multiple seeds, the inconsistency no longer appears, and the results align with the expected relationship between dimensionality and the required rank.
>
> We appreciate the reviewer for catching this issue; the revised table provides a more reliable and stable comparison.

---

### Meta-Review · Area_Chair_sMnX · 2026-01-07

**Summary:**

This paper proposes Dimension Domain Co-Decomposition (3D), a method for solving partial differential equations (PDEs) with Physics-Informed Neural Networks (PINNs). The method combines dimension decomposition, which factorizes the solution into dimension-wise components using a shared MLP for parameter efficiency, with domain decomposition, implemented via a Mixture-of-Experts (MoE). In addition, the paper introduces a new interpretability metric, Variable Interpretability (VI).

The scores were 4-4-4-0. Unfortunately, no discussion remains. The zero score may be extreme, but even disregarding this reviewer's comment, the overall impression is negative. Therefore, I do not recommend accepting this paper.

**Reviewer Concerns:**

The reviewers raised several concerns regarding both theoretical justification and model design. In particular, the method is considered a mere combination of existing techniques, such as dimension decomposition, parameter sharing, and MoE without clearly articulating a core technical insight or theoretical motivation for their integration. Regarding Variable Interpretability (VI), no theoretical justification is provided. Also, it was pointed out that the evaluation lacks comparisons with state-of-the-art PDE solvers. Although a high-dimensional Poisson problem is tested, the benchmarks remain relatively simple, and the reported accuracy and efficiency gains over standard PINNs are modest. Overall, the contribution is viewed as limited, leading some reviewers to question its novelty and practical impact.

To address the concerns, the authors respond that, for example, the proposed approach requires no manual domain pre-partitioning, incorporates dimension-wise interpretability as a core design principle, and enables automatic specialization of experts. Unfortunately, the discussion was not available, so it is unclear whether these comments addressed the reviewers' concerns or not. However, given that one reviewer also pointed out the paper's overall immaturity, it is unlikely that the concerns have been fully addressed.

**Reviewer Scores:**

The scores were 4-4-4-0. One reviewer mentioned a possibility for raising the score. If this reviewer raised the score, the scores could be 6-4-4-0. Even ignoring the extreme value of 0, the expected averaged score is less than 5.

---

### Decision · Program_Chairs · 2026-01-26

Reject